

# Investigating toxicity changes of cross-community redditors from 2 billion posts and comments

Hind Almerekhi[1], Haewoon Kwak[2] and Bernard J. Jansen[3]

[1] Hamad Bin Khalifa University, Doha, Qatar
[2] Singapore Management University, Singapore, Singapore
[3] Qatar Computing Research Institute, HBKU, Doha, Qatar

## ABSTRACT

This research investigates changes in online behavior of users who publish in multiple communities on Reddit by measuring their toxicity at two levels. With the aid of crowdsourcing, we built a labeled dataset of 10,083 Reddit comments, then used the dataset to train and fine-tune a Bidirectional Encoder Representations from Transformers (BERT) neural network model. The model predicted the toxicity levels of 87,376,912 posts from 577,835 users and 2,205,581,786 comments from 890,913 users on Reddit over 16 years, from 2005 to 2020. This study utilized the toxicity levels of user content to identify toxicity changes by the user within the same community, across multiple communities, and over time. As for the toxicity detection performance, the BERT model achieved a 91.27% classification accuracy and an area under the receiver operating characteristic curve (AUC) score of 0.963 and outperformed several baseline machine learning and neural network models. The user behavior toxicity analysis showed that 16.11% of users publish toxic posts, and 13.28% of users publish toxic comments. However, results showed that 30.68% of users publishing posts and 81.67% of users publishing comments exhibit changes in their toxicity across different communities, indicating that users adapt their behavior to the communities' norms. Furthermore, time series analysis with the Granger causality test of the volume of links and toxicity in user content showed that toxic comments are Granger caused by links in comments.

Corresponding author
Hind Almerekhi,
h.almerekhi@gmail.com,
hialmerekhi@hbku.edu.qa

## INTRODUCTION

Online social media platforms enable users to communicate with each other in various ways, like sharing and publishing different types of content (*Mondal, Silva & Benevenuto, 2017*). Unfortunately, the rapid growth of online communication on social media platforms has caused an explosion of malicious content in the form of harassment, profanity, and cyberbullying (*Hu et al., 2013*). A survey by Pew Research Center (*Vogels, 2020*) showed that 41% out of 10,093 American adults were personally harassed online, and 25% experienced severe forms of harassment. Moreover, 55% of the survey participants considered online

harassment a major problem. This concern was also shared by online moderators (*Cheng, Danescu-Niculescu-Mizil & Leskovec, 2015*), noting that posts and comments on many social media platforms can easily take a dark turn and become *toxic*. Therefore, there is a need for solutions that identify toxic content and limit its presence on online platforms.

One challenge with studying online toxicity is the multitude of forms it takes (*Davidson et al., 2017*). These forms include hate speech, which refers to offensive content that targets a specific trait in a group of people (*Silva et al., 2016*); harassment, which occurs when a user deliberately aggravates other users online (*Cheng, Danescu-Niculescu-Mizil & Leskovec, 2015*); and cyberbullying, which means targeting and intimidating victims through online communication (*Bowler, Knobel & Mattern, 2015*).

The previous classifications of toxicity forms show that toxic content often contains insults, threats, and offensive language, which, in turn, contaminate online platforms (*Mondal, Silva & Benevenuto, 2017*) by preventing users from engaging in discussions or pushing them to leave (*Newell et al., 2016*). Thus, several online platforms have implemented prevention mechanisms, such as blocklists (*Jhaver et al., 2018*), that block specific accounts from interacting with users. Other approaches to preventing toxicity include deploying human moderators and bots to remove toxic content (*Chandrasekharan et al., 2018*). These efforts, however, are not scalable enough to curtail the rapid growth of toxic content on online platforms (*Davidson et al., 2017*). There is also the psychological distress associated with exposing human moderators to firsthand accounts of toxicity (*Rodriguez & Rojas-Galeano, 2018*). These challenges call for developing effective automatic or semiautomatic solutions to detect toxicity from a large stream of content on online platforms.

Users of social media platforms have various reasons for spreading harmful content, like personal or social gain (*Squicciarini, Dupont & Chen, 2014*). Studies show that publishing toxic content (*i.e.,* toxic behavior) is contagious (*Tsikerdekis & Zeadally, 2014*; *Rodriguez & Rojas-Galeano, 2018*); the malicious behavior of users can influence non-malicious users and leads them to misbehave, which affects the overall well-being of online communities. As an example of toxic behavior (*Alfonso & Morris, 2013*), one Reddit user named *Violentacrez* created several communities on controversial topics such as gore, and his followers mimicked this behavior by creating communities with highly offensive content as well. This influence-based culture (*Johnson, 2018*) that users follow in online communities motivates studies like the current study to investigate the problem of toxic online posting and commenting behavior. Fueled by cases reported in *Alfonso & Morris (2013)*, this study focuses on the toxic behavior of users on Reddit. In particular, this research investigates the toxic cross-community behavior of users, which refers to publishing toxic content in more than one community.

This study argues that the toxicity of users' content may *change* based on the environment (*i.e.,* community) in which they participate. Therefore, the focus is to investigate changes in toxicity in two types of content that Reddit describes as follows:

- *Post*: is the top-level submission of a user that can be either a post, link, video, image, or poll.

- *Comment*: is the response of another user or the poster to a post or a comment.

This study uses an extensive collection of more than 2.293 billion published content, including 87 million posts and 2.205 billion comments, from more than 1.2 million unique users who published content in more than 107,000 unique subreddits from June 2005 to April 2020.

## LITERATURE REVIEW

Studies of online toxicity typically tackle the problem of hate from three main perspectives: (1) toxic behavior characterization, (2) toxic behavior detection, and (3) toxic behavior in online communities.

### Toxic behavior characterization

Investigating human behavior is essential for organizations that rely on users to drive business, understand the dynamics of online communities, and prevent hate (*Mathew et al., 2020*; *Yin & Zubiaga, 2021*). Negative behavior of humans in online spaces involves a lack of inhibition, including online aggressiveness that would not exist in similar situations offline (*Lapidot-Lefler & Barak, 2012*). *Suler (2004)* introduced the phrase "toxic disinhibition" and defined it as the inhibition loss of users who act violently online, which holds no benefits and leads users to violate conventional coexistence rules (*Suler, 2004*). A typical form of toxic disinhibition is flaming behavior, which involves using hostile expressions to refer to other users in online communication. Textual features of flaming behavior include harsh language, negative connotations, sexual harassment, and disrespectful expressions (*Pelicon et al., 2021*). The definition of toxic disinhibition, or toxic behavior, varies based on the users, the communities, and the types of interactions (*Shores et al., 2014*). For instance, toxic behavior can consist of cyberbullying and deviance between players in massively multiplayer online games (MMOGs) (*Shores et al., 2014*; *Kordyaka, Jahn & Niehaves, 2020*) or incivility between social media platform users (*Maity et al., 2018*; *Pronoza et al., 2021*), among other scenarios. In this work, we define *toxic behavior* in online communities as disseminating (*i.e.,* posting) toxic content with hateful, insulting, threatening, racist, bullying, and vulgar language (*Mohan et al., 2017*).

### Toxic behavior detection

There are two known methods for detecting toxic behavior on online platforms. The first relies on social network analysis (SNA) (*Wang & Lee, 2021*). The study of *Singh, Thapar & Bagga (2020)* exemplifies SNA usage and content-based analysis to detect cyberbullying (a form of toxic behavior) on Twitter. The study investigates the Momo Challenge, a fake challenge spread on Facebook and other social media platforms to entice younger users to commit violent acts. Researchers collected incidents related to the challenge by tracking 5,615 users' network graphs and 7,384 tweets using the Momo Challenge hashtag and relevant keywords. Findings showed that a small number of users employed keywords related to the Momo Challenge to cause cyberbullying events, whereas the majority used the keywords to warn other users about the dangerous challenge. Techniques involving SNA are suitable for detecting toxic behavior patterns and targeted attacks like cyberbullying.

However, these techniques must analyze the involved users' social profiles or relations to detect toxic behavior, which may not be available on platforms other than social media websites. Therefore, the second and most common method avoids this limitation by detecting toxic behavior in user-generated content (*Djuric et al., 2015*).

Analyzing user-generated content involves detecting toxicity; this is a heavily investigated problem (*Davidson et al., 2017*; *Ashraf, Zubiaga & Gelbukh, 2021*; *Obadimu et al., 2021*). To detect toxic content, some studies (*Nobata et al., 2016*) build machine learning models that combine various semantic and syntactic features. At the same time, other studies use deep multitask learning (MTL) neural networks with word2vec and pretrained GloVe embedding features (*Kapil & Ekbal, 2020*; *Sazzed, 2021*). As for open-source solutions, Google offers the Perspective API (*Georgakopoulos et al., 2018*; *Mittos et al., 2020*), which allows users to score comments based on their perceived toxicity (*Carton, Mei & Resnick, 2020*). The API uses pretrained machine learning models on crowdsourced labels to identify toxicity and improve online conversations (*Perspective, 2017*).

By using the outcomes of previous studies (*Wulczyn, Thain & Dixon, 2017*; *Georgakopoulos et al., 2018*), this work evaluates the performance of classical machine learning models (*Davidson et al., 2017*) and neural network models (*Del Vigna et al., 2017*) to detect toxicity at two levels from user content.

## Toxic behavior in online communities

Online platforms continuously strive to improve user engagement through various forms of interaction. Websites such as Facebook and Reddit offer users the freedom to create communities of their own volition to interact with similar groups of users (*Johnson, 2018*). Despite the great interest in promoting healthy interactions among users in online communities, platforms struggle with the toxic behavior of some unsolicited users (*Shen & Rose, 2019*). This problem was evident on Reddit (*Almerekhi, Kwak & Jansen, 2020*; *Massanari, 2017*), where *Chandrasekharan et al. (2017a)* found that some of the topics discussed by communities were incredibly toxic, leading to the 2015 ban of two hateful communities due to users' fears that these groups would infect other communities. The study found that the ban successfully prevented hate groups from spreading their toxicity to other communities. Nevertheless, this ban broke one of Reddit's core self-moderation policies, which exasperated users who sought total freedom on Reddit.

In a similar vein, *Mohan et al. (2017)* investigated the impact of toxic behavior on the health of communities. The study defines health as user engagement relative to community size and measures toxicity with a commercial lexicon from Community Sift to filter toxic words and phrases. By analyzing 180 communities, the study found a high negative correlation between community health and toxicity. Additionally, the study showed that communities require stable toxicity levels to grow in size without declining health. Despite these findings, the study did not consider users when investigating toxicity and viewed communities through content, not content creators.

As for cases in which toxic behavior arises between communities on different platforms, *Chandrasekharan et al. (2017b)* proposed a solution that relies on building a Bag of Communities (BoC). The research identified the abusive behavior of users in nine

communities from Reddit, 4chan, MetaFilter, and Voat. By computing cross-platform post similarity, the proposed model achieved 75% accuracy without any training data from the target platform. Moreover, the BoC model can achieve an accuracy of 91% after seeing 100,000 human-moderated posts, which outperforms other domain-specific approaches. However, the study focused on cross-platform abusive behavior through content analysis without accounting for the users who behaved in an abusive or toxic manner.

### Research questions

Given the literature review discussed earlier, this study aims to answer the following research questions:

**RQ1:** *How can the toxicity levels of users' content and users across different communities be detected?*

**RQ2:** *Does the toxicity of users' behavior change (a) across different communities or (b) within the same community?*

**RQ3:** *Does the toxicity of users change over time across different communities?*

## METHODOLOGY

In our study, investigating toxic behavior on Reddit requires a rigorous process that starts with obtaining the corpus from Reddit to detect toxicity and ends with finding insights from users' behavior.

### Data collection site

Reddit is an online community with over 2.8 million sub-communities (https://www.oberlo.com/blog/reddit-statistics; retrieved on 25 May 2022) that cover various topics from news to entertainment, incorporating a mix of cultures (*Massanari, 2017*). Those sub-communities are commonly known as "subreddits", denoted with the prefix "r/". The main activities that registered users (often called Redditors) perform include (a) creating subreddits, (b) submitting posts (*i.e.,* sharing content in the community), (c) commenting on the posts of others, and (d) replying to comments in discussion threads (*Choi et al., 2015*; *Kumar et al., 2018*).

Figure 1 shows a post along with responses (*i.e.,* comments) from "r/science." We can see in the figure that despite the strict moderation in this community, comments like "Statistically, pretty people are stupid" might be perceived by some Redditors as toxic.

### Obtain corpus

In this study, we targeted users in the top 100 subreddits (ranked by subscriber count). These subreddits account for a major proportion of Reddit's overall activity because they attract the largest number of active users (*Choi et al., 2015*). First, we compiled a list of subreddits using two popular external services: RedditList (http://redditlist.com; retrieved on Aug. 29, 2017), and RedditMetrics (https://frontpagemetrics.com/top; retrieved on Aug. 29, 2017). We used both websites to curate a list of the top 100 largest safe-for-work subreddits based on subscriber count. In Appendix A, Tables A1 and A2 show the top 100 subreddits sorted by total subscribers. Note that while the subreddit r/announcements

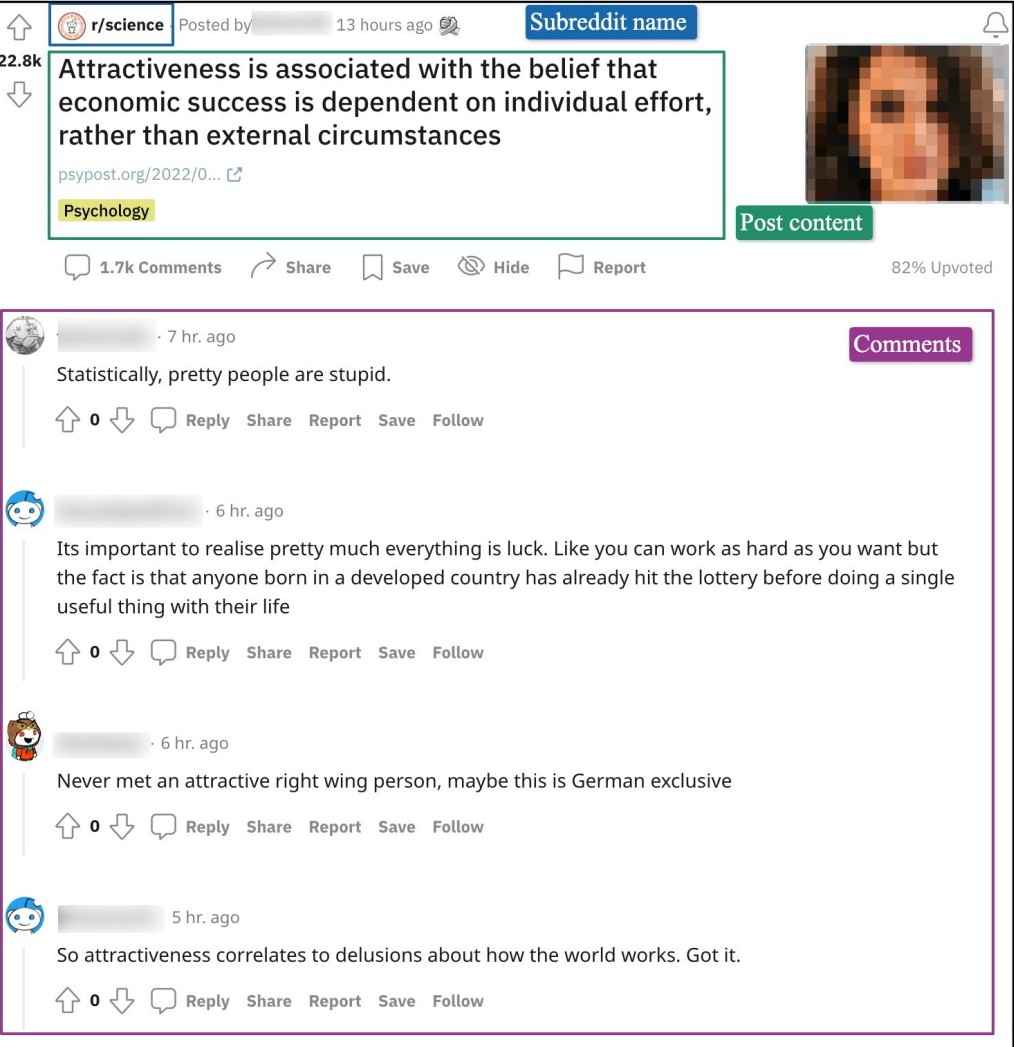

**Figure 1** A Reddit post from the subreddit "r/science" with its associated discussion threads.

holds the highest number of subscribers, we removed it from our list because it did not serve our study purpose, as most of the users of the subreddit use it to consume content. Moreover, the subscriber values were retrieved on Aug. 29, 2017, while the total posts and comments are from June 2005 to April 2020. While there might be some differences among the top 100 subreddit rankings over the years, we used the list from 2017 because our study aims to understand the toxic behavior of a subgroup of users across multiple communities over time. Therefore, this specific list does not harm the research goal. Instead, the list brings unique opportunities for tracking the toxic behavior of a user subgroup over time.

Since this study focuses on users and their content, we cleaned our user collection by dropping any deleted or removed users (*i.e.,* users with a removed or deleted "author" field). This process yielded 3,208,002 users who posted and 5,036,095 users who commented from 2005 through August 2017 within the top 100 subreddits.

Additionally, we excluded bot users (*i.e.,* automated accounts) to avoid potential biases in the subsequent analysis by using a publicly available list of 320 bots on Reddit (https://www.reddit.com/r/autowikibot/wiki/redditbots; retrieved on May 13, 2019). Since the available bot list is outdated, it potentially misses newer bot accounts. Thus, for this work, we used the Pushshift API (https://pushshift.io; retrieved on May 22, 2019) to retrieve a list of accounts with a minimal comment reply delay. Setting the comment reply delay to 30 s allowed us to find more bot accounts that quickly reply to other users. We removed additional bot accounts by combining the bot list and Pushshift API list. When conducting this study, we found 37 bot accounts that produce around 2% of automated content. The massive volume of bot-generated content reaffirms the importance of removing bots in the data-cleaning phase of this study.

Since our research focuses on the toxic cross-community behavior of users, each user must participate in at least two different subreddits from the list of top 100 subreddits. Therefore, we filtered our original list of users to remove users who only participate in a single subreddit. This filtering process returned 577,835 users who posted (18% of the 3,208,002 users) and 890,913 users who left comments (17.7% of the 5,036,095). Furthermore, the intersection of these user lists yielded 241,138 users that performed both acts of posting and commenting. Overall, our dataset has 1,227,610 unique users who post and comment on Reddit. Lastly, we built our final collection of posts and comments with the Pushshift API (*Baumgartner, 2017*) by extracting user content from all subreddits. In other words, we started with a group of users participating in multiple communities, extracted their content from the top 100 subreddits, and then extracted their content from all other subreddits. As a result, we extracted 87,376,912 posts from 76,650 subreddits and 2,205,581,786 comments from 79,076 subreddits. To summarize, our collection has 2,292,958,698 posts and comments from 107,394 unique subreddits made by a group of cross-community users from June 2005 to April 2020.

Figure 2 shows the Cumulative Distribution Function (CDF) of the number of subreddits where a user left posts (A) and comments (B). Once a user participates in multiple subreddits (we already removed users who participated in a single subreddit, which is around 80% of users), the number of subreddits they participate in quickly grows. Findings from Fig. 2 indicate that users who participate in less than 10 subreddits are more than 80% and 50% in terms of posting and commenting, respectively. Also, participation through commenting seemed to be easier than posting; in Fig. 2, users who left comments in more than 20 subreddits are higher than 20%. In summary, the user collection in this study captured a substantial number of cross-community interactions and thus was appropriate for examining toxic behavior across multiple communities.

## Label dataset

To investigate the toxicity of users, we required a reliable machine learning model to detect the toxicity of user content. However, before building the detection model, we first created a set of relevance judgments (*i.e.,* labels) that determine if a particular comment is toxic or not. Before conducting this study, we found a few publicly available toxicity detection datasets, such as the Wikipedia comments training set that Google's Jigsaw/Conversation AI released

(A)                                  (B)

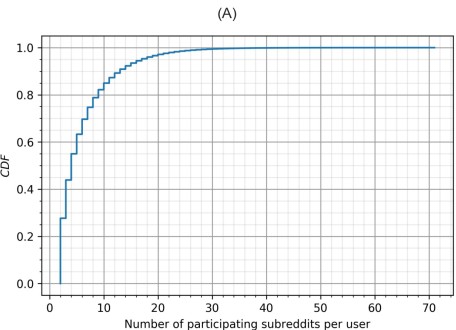
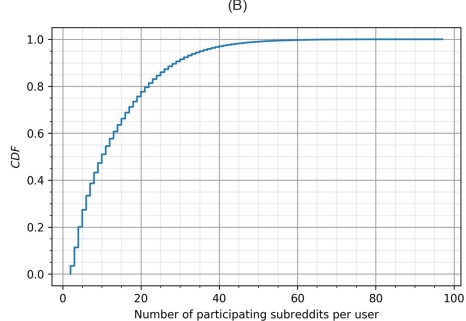

**Figure 2** Cumulative distribution function of the participating subreddits per user in (A) posts and (B) comments.

on Kaggle (https://www.kaggle.com/c/jigsaw-toxic-comment-classification-challenge; retrieved on Feb. 13, 2018). However, we found that the dataset targets a classification problem that does not align well with our problem. Therefore, we created a labeled collection with data from our collection of comments.

To conduct our labeling experiment, we randomly sampled 10,100 comments from r/AskReddit, one of the largest subreddits in our collection. First, we used 100 comments to conduct a pilot study, after which we made minor modifications to the labeling task. Then, we proceeded with the remaining 10,000 comments to conduct the complete labeling task. We selected 10,000 comments to ensure that we had both a reasonably-sized labeled collection for prediction experiments and a manageable labeling job for crowdsourcing. For labeling, we recruited crowd workers from Appen (https://appen.com; retrieved on Jun. 10, 2022) (formerly known as Figure Eight). Appen is a widely used crowdsourcing platform; it enables customers to control the quality of the obtained labels from labelers based on their past jobs. In addition to the various means of conducting controlled experiments, this quality control makes Appen a favorable choice compared to other crowdsourcing platforms.

We designed a labeling job by asking workers to label a given comment as either toxic or nontoxic according to the definition of a toxic comment in the Perspective API (*Perspective, 2017*). If a comment was toxic, we asked annotators to rate its toxicity on a scale of two, as either (1) slightly toxic or (2) highly toxic. To avoid introducing any bias to the labeling task, we intentionally avoided defining what we consider highly toxic and slightly toxic and relied only on crowd workers' judgment on what the majority of annotators perceive as the correct label (*Vaidya, Mai & Ning, 2020*; *Hanu, Thewlis & Haco, 2021*). Nonetheless, we understand that toxicity is highly subjective, and different groups of workers might have varying opinions on what is considered highly or slightly toxic (*Zhao, Zhang & Hopfgartner, 2022*). Therefore, annotators had to pass a test by answering eight test questions before labeling to ensure the quality of their work.

Additionally, we used 70 test questions for quality control during the labeling process. The test questions for the labeling task were comments that were undoubtedly toxic or nontoxic. Such questions were given to annotators at random intervals to detect spammers

and ensure the quality of the obtained labels. One of the labeling task settings was to get the agreement of three annotators for each comment, meaning that at least two of the three annotators had to agree for a comment to be labeled. Moreover, crowd workers were required to spend a minimum of 30 s on each page while labeling and had a minimum labeling accuracy of 80% based on prior experience with Appen. Unfortunately, Appen does not provide any demographic statistics about the participating workers. Therefore, we only know that our workers had an accuracy ≥ 80% and could understand English.

To assist workers with the labeling task, we provided a link to the discussion thread of the comment on Reddit to establish the context of the discussion. Regarding the job design, we split the job into 2,000 pages, each with five rows. As for costs, we paid USD 490.2 for the pilot (USD 9.48) and the complete labeling task (USD 471.24). The details of the labeling job are in Appendix B, where we show the labeling job instructions in Fig. B1 and the labeling job test questions in Fig. B2.

To assess the quality of the obtained labels from crowdsourcing, we computed a set of reliability measures (http://mreliability.jmgirard.com; retrieved on Dec. 15, 2017) based on the labels from every worker for all classes. In this context, we define reliability as the proportion of perceived non-chance agreement to potential non-chance agreement (*Gwet, 2014*). To compute reliability, we used Eq. (1), where ($P_o$) is the percent of observed agreement and ($P_c$) is the percent of chance agreement. Generally, there are three main approaches to estimate the agreement by chance between two or more annotators across bifurcated categories: (a) individual-distribution-based approach like Cohen's Kappa (*Zhao, Liu & Deng, 2013*), (b) category-based approach such as Bennett et al.'s S score (*Bennett, Alpert & Goldstein, 1954*), and (c) average-distribution-based approaches, such as the ones adopted by Gwet's gamma coefficient (*Gwet, 2014*).

$$Reliability = \frac{P_o - P_c}{1 - P_c}. \tag{1}$$

Some of the reliability measures and results are in Table 1. Each measure takes into account various parameters based on its specific implementation. It is worth noting that while the observed agreement is high, Fleiss's Kappa is relatively low; this problem was heavily discussed by *Feinstein & Cicchetti (1990)* and *Salminen et al. (2019)*, with the low Kappa score attributed to the class imbalance. On the other hand, Gwet's gamma (*Gwet, 2014*) and other average distribution approaches handle such class imbalance differently and outperform the Kappa statistic measurements. Given the class imbalance in our dataset, representative of the online discussion community, we use Gwet's measure in our research. Lastly, we sampled 100 random comments from our training set and manually labeled them for their toxicity. Then, we measured the agreement between our labels and crowd workers. Our findings showed that there is an agreement of 92%, which means that crowd labels are suitable for training machine learning models.

Out of the 10,100 labeled comments, we removed three duplicate comments and 14 comments modified by moderators (*i.e.,* the actual content of the comments was either removed or heavily edited by automatic bots or moderators). Of the 10,083 labeled comments, 86.81% were labeled as nontoxic, while the remaining 13.82% were labeled

**Table 1** **Various reliability measure results on the relevance judgments obtained from Appen.** ICC is Intraclass Correlation Coefficient.

| Measure | Obtained value |
|---|---|
| Observed agreement | 0.848 |
| Gwet's gamma coefficientcitep (*Gwet, 2014*) | 0.697 |
| Average rater consistency ICC | 0.738 |
| Average rater agreement ICC | 0.737 |
| Bennett et al. Score (*Bennett, Alpert & Goldstein, 1954*) | 0.697 |
| Conger's kappa coefficient | 0.483 |
| Scott's pi, Krippendorff's alpha & Fleiss's kappa coefficient | 0.482 |

**Table 2** **Examples of comments from every class as labeled by Appen workers.**

| Comment | Class |
|---|---|
| ...Ex. Megan Fox is less hot to me because I have heard she is a fucking cunt. | Highly toxic |
| Fuck all of you Jersey haters. I harbor no hate for any states, except those in the south... | |
| Fuck that. I love seeing bitches lose their shit and go full retard on each other. | |
| Fuck I hate people that delete or not have a facebook account... | Slightly toxic |
| She always does this hideous (like her face) fake laugh and says that I'm such a dumb blonde... | |
| ...Other people do well so they must be evil rather than hard working smart sons-abitches, huh? | |
| I have pretty extreme ADHD and I do it to stay focused... | Nontoxic |
| I feel you on this one...with the exception of hard boiled egg in ramen... | |
| What I am saying is that the word "talent" is not that well defined... | |

toxic (2.78% highly toxic and 11.04% slightly toxic). Some of the labeled comments from our collection are in Table 2. Due to space limitations, we only show relevant portions of the comments in the examples from Table 2.

## Build models

An integral part of our methodology involves building a robust prediction model that can classify posts and comments based on the crowdsourced labels into three classes, which are (a) nontoxic, (b) slightly toxic, and (c) highly toxic. The following subsections describe the features extracted from the labeled dataset and the classification models we evaluated to predict toxicity at different levels.

### *Extracting features*

For the prediction task, we utilized various features that characterize the semantic properties of text. First, we examined n-gram features at different configurations; then, we extracted an advanced set of features based on word embeddings, followed by a set of NLP-based features derived from the comments text. The following subsections explain each of the feature categories in more detail.

*N-gram features.* Before computing all the features, we cleaned the collection by removing new lines, URLs, and Reddit-specific (https://github.com/LoLei/redditcleaner; retrieved on Oct. 16, 2021) symbols like bullet points, quotes, strikethrough, spoilers, and coding parts from Reddit text. Furthermore, we split the collection text into tokens based on spaces and normalized all capital letters to lowercase letters. Then, we extracted 3,000 feature vectors from multiple variations of n-gram representations, including unigram features, bigram features, TF-IDF features, and n-gram features with a token range from three to five (*Yin et al., 2009*).

*Word embedding features.* For embedding features, we created vectors in a low-dimensional space using a distributed memory model from the vector representations of the cleaned tokens in the collection (*Le & Mikolov, 2014*). So, we used Python's Gensim library (https://radimrehurek.com/gensim/; retrieved on May 20, 2018) to build a skip-gram model and train the embeddings using a window size of 10, hierarchical softmax, and negative sampling of 10 noisy words. Then, we used the model to generate 300 word2vec feature vectors. Lastly, we trained another skip-gram model of window size 15, negative sampling of seven noisy words, and a learning rate of 0.03 to represent sentences as 300 doc2vec features.

*NLP-based features.* In addition to the previous features, we computed 37 shallow features based on natural language processing (NLP) techniques. Table 3 shows a summary of the list of features divided by the type of calculations we performed to obtain such features. We found that all the NLP-based features typically involve counting tokens or measuring the ratios of counts. Some of the features in Table 3 are adopted from *Salminen et al. (2018)*, where they used similar features to identify hateful posts on Facebook and YouTube.

## Classify content
### Classification based on classical machine learning
The classification approach considered several issues that persisted in the collection, such as the skewness of the classes. Since the labeled collection is highly skewed (86.81% of the comments are non-toxic), we had to address the class imbalance issue. One way to address this issue is to apply the Synthetic Minority Over-sampling Technique (SMOTE) and Tomek links (an under-sampling technique) (*Batista, Prati & Monard, 2004*). SMOTE performs over-sampling, and links are cleaned by under-sampling using Tomek links. The next step in feature-based classification was to apply feature transformation by following a simple min-max scaling approach to normalize all features. Then, we performed feature selection to reduce the dimensionality of large feature vectors like n-grams. The last step in the classification procedure involved performing a grid search for parameter tuning followed by repeated stratified cross-validation over five folds.

### Classification based on neural networks
Recurrent neural networks (RNNs) are artificial neural networks designed to process a sequence of inputs with temporal characteristics. One famous example is the long short-term memory (LSTM) neural network consisting of an input gate, a cell, a forget

**Table 3  The list of 37 NLP-based features split into two categories based on the type of computation.**

| Feature types | List of features |
|---|---|
| **Counts**<br>**20 feature** | Characters (text length), words, capitals, nouns, verbs, adjectives, stop words, punctuations, periods, quotes, unknown words, discourse connectives, politeness words, rudeness words, single tokens, repeated punctuations, unique words, profane words, modal words, non alpha-numeric characters |
| **Ratios (a:b)**<br>**17 features** | $\begin{cases} \text{a = counts of words, capitals, stop words, unique words, punctuations, nouns, verbs, adjectives} \\ \text{b = text length} \end{cases}$<br>$\begin{cases} \text{a = counts of capitals, characters (without spaces), stop words, unique words, punctuations,} \\ \text{profane words, nouns, verbs, adjectives} \\ \text{b = count of words} \end{cases}$ |

gate, and an output gate. Another neural network type is the convolution neural network (CNN or ConvNet), which is commonly used to analyze images. However, a CNN can also be employed in other applications to detect data patterns (*Johnson & Zhang, 2016*).

In natural language processing, one of the most prominent deep learning networks is the transformer (*Devlin et al., 2019*), which handles sequences of ordered data to solve problems like machine translation. The difference between RNNs and transformers is that the latter do not require data to be processed in order. With this advantage, transformer models can be faster than RNNs during the training phase due to the parallelization of the data processing phase (*Vaswani et al., 2017*). A well-known transformer model is BERT, which consists of a multilayer bidirectional transformer encoder (*Devlin et al., 2019*).

## Explore results
### Toxicity judgments of user content
To determine user toxicity, we compute the percentages of highly toxic content and slightly toxic content. Combining these provides a general judgment of a user's toxic behavior, regardless of the toxicity level. Furthermore, combining toxicity levels compensates for the skewness of the dataset by increasing the amount of data that represents what is considered toxic. For instance, if a user *u* creates three posts, one labeled highly toxic, one slightly toxic, and one nontoxic, this user *u* is 67% toxic. Then, we use percentiles to describe users based on the proportion of toxicity in their generated content. The quartile values, which include the 25th, 50th, 75th, and 100th percentiles, capture the distribution of users based on the toxicity proportions in their content. For instance, if the 25th percentile is 10, it means that 25% of the time, the toxicity proportions in users' posts are below 10.

### Toxicity changes of users in subreddits
Prior studies of the behavior of online users found that temporal features can help characterize undesirable behavior like trolling (*Cheng, Danescu-Niculescu-Mizil & Leskovec, 2015*). Thus, investigating the temporal aspects of toxic behavior is an interesting problem from the perspective of *Mathew et al. (2019)*; *Kaakinen, Oksanen & Rsnen (2018)*, and *Cheng, Danescu-Niculescu-Mizil & Leskovec (2015)*. Toward this goal, we studied toxicity changes by computing the toxicity difference of user content across all subreddits. In Eq. (2), we illustrate how to calculate the change ($\Delta$) of toxicity per user. To get the percentage of toxic content, we combined the counts of highly toxic and slightly toxic content that

users made within a particular subreddit. Then, we divided this figure by the total number of content items posted by these users within that same subreddit. Next, we computed the difference by subtracting the highest toxicity proportion from each user's lowest toxicity within a particular year. Finally, from Eq. (2), we computed the differences in users' content from posts and comments over the years.

$$\Delta_u = \max_s \left( \frac{N(c_{toxic})_u^s}{N(c_{total})_u^s} \right) - \min_s \left( \frac{N(c_{toxic})_u^s}{N(c_{total})_u^s} \right). \tag{2}$$

$N(c_{toxic})_u^s$ is the number of toxic content (comments or posts) by user $u$ in subreddit $s$, and $N(c_{total})_u^s$ is total number of content (comments or posts) by user $u$ in subreddit $s$.

For instance, if a subreddit $s$ has a total of 20,000 posts, and a user $u$ posted 1,000 slightly toxic and 800 highly toxic posts, we add both highly and slightly toxic posts to get a total of 1,800 toxic posts. Then, we divide the total number of toxic posts by the total number of posts in $s$ to get a toxicity percentage of 0.09. With this procedure, we continue to get all the toxicity percentages of $u$ to calculate $\Delta$ like Eq. (2). Obtaining the toxicity percentage of all users within subreddits in the posts and comments collections is necessary for subsequent analysis in our study.

### Link analysis of user content

Another way to investigate user behavior is by looking at the links in their content. Since our collection is rich in metadata, we extracted the URL field that includes the link (if any) that accompanies a post. Then, we searched for all the URLs from comments text to build another version of our dataset that includes an ID, a time stamp, and URLs. We included the time stamp in this version of the dataset to conduct some statistical hypothesis tests, such as the Granger causality test (*Mukherjee & Jansen, 2017*) to identify relationships between URLs in user content and the toxicity of said content.

## FINDINGS

### Classification results

In the following subsections, we show the results of several experiments that build and evaluate machine learning models for detecting toxicity at different levels.

### Classical classification models

For this part of the analysis, we computed features to build and tune four widely used classic machine learning models: Logistic Regression, Random Forest, Decision Tree, and XGBoost, using the 10,083 comments from the labeled collection. We chose these four algorithms for their extensive usage in previous research on hate detection (*Badjatiya et al., 2017*; *Salminen et al., 2018*).

To handle class imbalance, we used SMOTE and Tomek Links on the training portion of the dataset (0.80 of the labeled collection). Then, we transformed features by scaling all values to a range between zero and one. Additionally, we used the Random Forest algorithm to perform the classification, ranking, and selection of features (*Vens & Costa, 2011*). The results depicted in Table 4 show the precision, recall, $F_1$, AUC, and classification accuracy

**Table 4  The classification performance of each feature category across four different classifiers.**

| Features | Logistic regression | | | | | Random forest | | | | | Decision tree | | | | | XGBoost | | | | |
|---|---|---|---|---|---|---|---|---|---|---|---|---|---|---|---|---|---|---|---|---|
| | *P* | *R* | *F₁* | *AUC* | *ACC.* | *P* | *R* | *F₁* | *AUC* | *ACC.* | *P* | *R* | *F₁* | *AUC* | *ACC.* | *P* | *R* | *F₁* | *AUC* | *ACC.* |
| Unigram | 0.649 | **0.692** | 0.668 | 0.878 | 85.1 | 0.653 | 0.443 | 0.479 | 0.882 | 84.1 | 0.618 | 0.606 | 0.611 | 0.763 | 85.2 | 0.715 | 0.653 | **0.679** | 0.913 | **88.1** |
| Bigram | 0.572 | 0.389 | 0.379 | 0.625 | 57.1 | 0.416 | 0.347 | 0.327 | 0.609 | 81.5 | 0.499 | 0.374 | 0.379 | 0.547 | 80.9 | 0.572 | 0.413 | 0.435 | 0.668 | 82.1 |
| N-gram (3-5) | 0.358 | 0.394 | 0.292 | 0.558 | 39.2 | 0.386 | 0.406 | 0.274 | 0.569 | 39.1 | 0.392 | 0.342 | 0.104 | 0.515 | 13.3 | 0.410 | 0.378 | 0.384 | 0.594 | 76.9 |
| TFIDF | 0.624 | 0.671 | 0.645 | 0.892 | 84.6 | 0.643 | 0.594 | 0.606 | 0.883 | 85.3 | 0.602 | 0.620 | 0.609 | 0.782 | 84.7 | 0.673 | 0.636 | 0.653 | 0.910 | 87.3 |
| NLP | 0.393 | 0.446 | 0.365 | 0.660 | 54.5 | 0.419 | 0.399 | 0.406 | 0.644 | 75.3 | 0.370 | 0.399 | 0.231 | 0.568 | 26.4 | 0.424 | 0.377 | 0.383 | 0.620 | 78.4 |
| Word2vec | 0.477 | 0.579 | 0.492 | 0.789 | 67.4 | 0.578 | 0.456 | 0.487 | 0.799 | 81.1 | 0.393 | 0.441 | 0.384 | 0.645 | 57.8 | 0.573 | 0.522 | 0.543 | 0.810 | 81.3 |
| Doc2vec | 0.498 | 0.593 | 0.523 | 0.828 | 73.8 | 0.570 | 0.531 | 0.548 | 0.822 | 81.9 | 0.426 | 0.474 | 0.434 | 0.652 | 66.4 | 0.561 | 0.551 | 0.556 | 0.815 | 81.3 |
| All Features | 0.610 | 0.641 | 0.624 | 0.893 | 84.1 | 0.662 | 0.509 | 0.552 | 0.884 | 84.9 | 0.592 | 0.578 | 0.584 | 0.735 | 84.6 | **0.732** | 0.636 | 0.671 | **0.924** | 87.8 |

**Notes.**

P, precision; R, recall.

of every classifier, where accuracy measures the number of correctly predicted data items divided by the total number of predicted items.

Findings from Table 4 show the best performing classification model is XGBoost, followed closely by Logistic Regression. As for the best features, the results show that models perform best on the unigram features. However, all the features combined through concatenation with XGBoost showed the highest precision and AUC scores at 0.732 and 0.924, respectively. On the other hand, with Logistic Regression, unigram features achieved a recall score of 0.692. As for the $F_1$ and accuracy, XGBoost achieved the highest scores of 0.679 and 88.1% on the unigram features. The grid search of XGBoost showed that the best learning rate is 0.3, and the best number of estimators is 300. Moreover, feature selection reduced the dimensionality of all the combined features from 12,637 to 1,406, where the top selected features belonged to unigram and word embedding feature categories. This outcome aligns with the prior work done by *Nobata et al. (2016)*, where their best-performing feature categories on all their datasets were the n-gram and distributional semantic features.

### Neural network models

Despite the outstanding performance of classic machine learning models, studies found that some neural network architectures can outperform classical machine learning models, especially with capturing long-range dependencies from textual data in hate detection problems (*Badjatiya et al., 2017*). Therefore, we chose to experiment with varying configurations of BERT as a basis for our trials with CNNs, RNNs (LSTM and biLSTM), and transformer networks.

- **CNN:** This model used a convolution layer along with global max pooling and batch normalization layers to normalize the layer dimensions and speed up the performance of the model (*Zhou et al., 2016*). The network deployed a learning rate of 0.00002. The optimizer was adam, and the maximum sequence length for tokenizing the training set was 384. The embedding features were from a pretrained BERT-medium model (https://huggingface.co/google/bert_uncased_L-8_H-512_A-8; retrieved on Oct. 20, 2021)

**Table 5** Performance of the neural network models in terms of the macro precision, recall, $F_1$, AUC, and accuracy scores.

| Neural network models | Evaluation metrics | | | | |
|---|---|---|---|---|---|
| | *Precision* | *Recall* | *Macro F1* | *AUC* | *Accuracy* |
| CNN | 0.6600 | 0.6172 | 0.6092 | 0.9231 | 87.12% |
| BiLSTM | 0.6222 | 0.6629 | 0.6216 | 0.9336 | 86.42% |
| CNN+LSTM | 0.6645 | 0.6966 | 0.6724 | 0.9380 | 87.51% |
| LSTM+CNN | 0.7431 | 0.7212 | 0.7261 | 0.9570 | 89.99% |
| fine-tuned BERT | 0.7930 | 0.8034 | 0.7952 | 0.9629 | 91.27% |

- **bidirectional LSTM:** The model used a bidirectional LSTM layer with embedding features from BERT-medium. Additionally, the model had average pooling layers with dense layers and was trained using the same learning rate and sequence size as the previous CNN model.
- **CNN+LSTM:** This model consisted of four channels with convolution layers, global max pooling, and batch normalization. In addition, the end of each channel has an LSTM layer. The final model consists of the combined channels with added drop-out layers. The same BERT-medium features were used in this model with the same configurations.
- **LSTM+CNN:** This model used a bidirectional LSTM layer followed by a series convolution, global max pooling, and batch normalization layers. Like the previous models, BERT-medium was used to obtain the feature vectors.
- **fine-tuned BERT:** This transformer model used the uncased (*i.e.,* lower case) base model of BERT, which consists of 12 layers (also called transformer blocks), 768 hidden layers, 12 attention heads, and 110 million parameters. To fine-tune the model, we used a custom focal loss (*Lin et al., 2017*) function with gamma $=2$ and alpha $=7$ to account for class imbalance. Additionally, we computed class weights from each class's distribution of data points and used them to improve the training performance. As for the learning rate, we set it to 0.00003 and used the Adam optimizer.

We evaluated the performance of the neural networks on the labeled training set, where 80% of the data was for training while the rest was for testing and validation purposes. Then, we used the dataset's testing portion to evaluate the models' performance by measuring macro precision, recall, $F_1$, AUC, and accuracy scores, as in Table 5. The findings show that, clearly, the fine-tuned BERT model outperforms all neural network models, as it achieves a precision score of 0.7930, recall of 0.8034, an $F_1$ score of 0.7952, AUC score of 0.9629, and accuracy score of 91.27%.

One drawback of using a BERT model is that training was relatively slow. However, this issue can be solved by adjusting the batch size configuration. Comparing the performance of neural network models in Table 5 with the classical classification models in Table 4, the results showed that the fine-tuned BERT model outperformed all of the other models. Even though the performance of the BERT model might look too good to believe, the neural-network-based model has shown its high performance in toxic-classification tasks. For example, Google Jigsaw hosted a toxicity detection challenge

**Table 6** Prediction results of the toxicity levels of the posts and comments of users.

| Class | Posts (%) | Comments (%) |
|---|---|---|
| Highly toxic | 1,794,115 (2.05%) | 133,588,229 (6.06%) |
| Slightly toxic | 6,364,092 (7.28%) | 254,531,824 (11.54%) |
| Nontoxic | 79,218,705 (90.66%) | 1,817,233,194 (82.39%) |

to classify toxic comments from Wikipedia in 2018, and the first ranked team reported an AUC score of 0.9890 (https://www.kaggle.com/c/jigsaw-toxic-comment-classification-challenge/discussion/52557; retrieved on Mar. 25, 2022).

In summary, our fine-tuned BERT model will be used to detect toxicity in the remainder of this study.

### *Transferability of models across subreddits*

To use the models to predict the toxicity of all the Reddit comments, we must ensure that the model trained by the data from r/AskReddit is transferrable to data from other subreddits (*Fortuna, Soler-Company & Wanner, 2021*). Toward this goal, we obtained a random sample of 1,000 comments from the remaining 99 subreddits (besides r/AskReddit). Then, we used crowdsourcing to label the comments for their level of toxicity. Finally, we used the same labeling job we described earlier to obtain the ground truth labels. Comparing the results from the prediction model and the crowdsourcing workers showed that the agreement between them was 94.2%, meaning that the model trained on r/AskReddit can be generalizable to other subreddits.

### Detecting and determining the toxicity of content and users

To answer RQ1 (*How can the toxicity levels of users' content and users across different communities be detected?*), we infer the toxicity of the entire posts and comments collection by using our fine-tuned BERT model. Post toxicity was detected by concatenating post titles and body sections (if they existed). As for comments, toxicity was predicted directly from the comment text. The results of running the prediction model on the entire collection of 87,376,912 posts and 2,205,581,786 comments are in Table 6. The results show that, collectively, 17.61% of the posts were toxic (*i.e.,* both highly toxic and slightly toxic) and that the remaining 82.39% were nontoxic.

After obtaining the toxicity levels of user posts, we applied our method to judge user toxicity and get the total number of users (and their toxicity percentages) in every quartile, as shown in Table 7. For users who leave posts, the table shows that, in the 25th percentile, 26.27% of users had toxicity proportions in the range (1%, 5%]. As for the 50th percentile, 25.56% of users had toxicity proportions that fell in the range (5%, 9%]. Subsequently, in the 75th percentile, the toxicity proportions for 24.91% of users were in the range (9%, 15%]. Additionally, Table 7 shows that, in the 100th percentile (*i.e.,* the maximum quartile), the toxicity proportions in 23.26% of users were in the range (15%, 100%]. Therefore, among the four quartiles, the 25th percentile had the largest number of users; the average toxicity was 3.40%.

**Table 7** Judgment of users' toxicity based on their predominant behavior. The judgments include the total number of users, their percentage (%), and the toxicity range of their posts and comments.

| Users judgment | Posts (%) - toxicity range | Comments (%) - toxicity range |
|---|---|---|
| 25th percentile | 124,056 (26.27%) - (1%, 5%] | 234,899 (26.55%) - (1%, 11%] |
| 50th percentile | 120,705 (25.56%) - (5%, 9%] | 209,133 (23.64%) - (11%, 16%] |
| 75th percentile | 117,651 (24.91%) - (9%, 15%] | 220,799 (24.96%) - (16%, 22%] |
| 100th percentile | 109,821 (23.26%) - (15%, 100%] | 219,891 (24.85%) - (22%, 100%] |

In the comments collection, for users in the 25th percentile, 26.55% had toxicity proportions in the range (1%, 11%]. Concerning the 50th percentile, 23.64% of users had toxicity proportions with a range of (11%, 16%]. The findings show that, in the 75th percentile, 24.96% of users had toxicity proportions with a range of (16%, 22%]. Lastly, our results show that, in the 100th percentile, 24.85% of users had toxicity proportions in the range (22%, 100%]. Simply put, in the 25th percentile of users who leave comments, about 27% of the users have an average toxicity of 7.77%.

## Changes in users' toxicity across communities

Since some users do not show consistent toxic (or nontoxic) behavior in their content, with RQ2 (*Does the toxicity of users' behavior change (a) across different communities or (b) within the same community?*), we examine the content-based changes in users' toxicity. Here, we check if a change (or a multitude of changes) in the toxicity of users occurs within the same subreddit or across different subreddits.

To examine toxicity changes in this study, we devised two different change conditions to look for in the users' collection. These conditions come from the two possible judgments of users' posting behavior based on their content's toxicity. Based on our methodology, we judge users based on their contributions as (1) toxic (slightly toxic and highly toxic) or (2) nontoxic. Based on these judgments, the conditions that we identified are as follows:

- **Condition 1:** Change in the toxicity of a user's contribution from nontoxic to toxic. (**NT → T**)
- **Condition 2:** Change in the toxicity of a user's contribution from toxic to nontoxic. (**T → NT**)

Additionally, this experiment checked whether the conditions were met within the same subreddit or across different subreddits. Given the criteria for investigating the change in toxicity, we examined the entire history of users' content in the posts and comments collections. First, we sorted content from oldest to newest based on time stamps. Then, we used the toxicity prediction labels that we obtained from our prediction model to check for the change conditions. For example, suppose the first (*i.e.*, oldest or earliest) post of a user is nontoxic and the subsequent (*i.e.*, following or newer) post is toxic. In that case, this user exhibits a change in toxicity due to condition 1. This experiment considers whether the change happened in the same subreddit or across multiple subreddits. Thus, we flagged every user based on the exhibited condition and identified the location of the change on Reddit. Subsequently, we used a majority voting mechanism to get the number of users

**Table 8  Total number of users (and their %) that satisfy the conditions and their locations on Reddit in the posts and comments collections.**

| Conditions | Submission users | | | Comment users | | |
| --- | --- | --- | --- | --- | --- | --- |
| | Total | Same subreddit | Multiple subreddits | Total | Same subreddit | Multiple subreddits |
| 1. NT →T | 23,946 (5.11%) | 3,668 (15.32%) | 16,377 (68.39%) | 10,500 (1.19%) | 2,608 (24.84%) | 6,412 (61.07%) |
| 2. T →NT | 10,123 (2.16%) | 1,468 (14.50%) | 4,631 (45.75%) | 7,891 (0.89%) | 1,987 (25.18%) | 4,572 (57.94%) |
| 1. NT →T & 2. T →NT | 435,000 (92.74%) | 15,869 (3.65%) | 419,131 (96.35%)* | 866,228 (97.92%) | 48,070 (5.55%) | 818,158 (94.45%)** |

**Notes.**
NT, Nontoxic; T, Toxic.
50.80% of 419,131* and 68.63% of 818,158** show change in the same and multiple subreddits.

that exhibited changes due to each condition and their locations on Reddit. We performed majority voting by getting a list of all the toxicity changes and their locations for every user. If most of the total changes for a user fall under a specific condition or location, then a user change is due to this particular condition. Moreover, any user who shows a single change in toxicity due to any condition was removed to avoid any issues that might arise by posting a single toxic post. For instance, if a user had two posts that showed a change in any of the conditions, we did not include it in the study. With this approach, we found that users can show at least two changes due to the one or two conditions at different locations on Reddit.

The majority voting technique allowed us to count users based on the overall change in their content. This approach resulted in 177,307 (30.68%) posting users and 727,587 (81.67%) commenting users that show changes due to conditions 1 or 2. To further analyze these users, the results depicted in Table 8 show the distribution of users who satisfy conditions within the same subreddit or across multiple subreddits in both posts and comments.

In Table 8, we measured the percentage of users that satisfy each condition along with the combined conditions and came up with compelling observations from each collection. For example, starting with the posts collection, we found that users show the most change due to conditions 1 and 2, where 96.35% of these users show a change across multiple subreddits. Additionally, 5.11% of these users show a change due to condition 1, where 68.39% of them show changes across multiple subreddits; similarly, the majority of users that satisfy condition 2 also show change across different subreddits. Specifically, 45.75% of these users show changes over multiple communities, meaning that most of the changes in toxicity among users that post occur in different communities.

As for the comments collection, just like the posts collection, users show the most change due to conditions 1 and 2, where 1.19% of these users show a change within the same subreddit. Moreover, 0.89% of these users show change due to condition 2, where 57.94% of them show changes across different subreddits. In other words, most commenting users show changes in their toxicity across different subreddits. So, concerning RQ2, findings show that engaging with multiple communities can cause users to exhibit changes in their toxic posting behavior.

To further illustrate the changes in toxic behavior, we count the total number of changes per user without considering the majority voting technique we used in Table 8. Then,

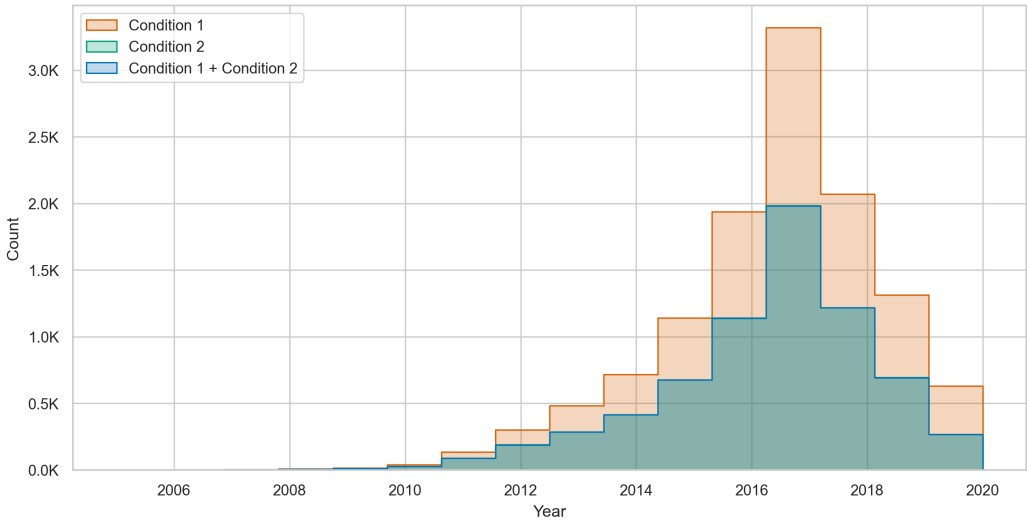

**Figure 3** The count of toxicity changes over time in posting users from condition 1 (NT → T), condition 2 (T → NT), and both conditions combined.

we plot a histogram of the counts of changes that occur due to condition 1, condition 2, and both conditions combined. The histogram in Fig. 3 shows that over time, most of the changes are in condition 1, where posting users change their behavior from nontoxic to toxic.

Similarly, Fig. 4 shows that for commenting users, most of the changes occur due to condition 1. However, the gap between the counts of changes due to varying conditions is smaller in commenting users. Furthermore, we found that posting users can show up to 37,800 changes in toxicity while commenting users can show up to 295,912 changes due to both conditions. These high numbers of changes suggest that users change their toxicity when their volume of contributions increases. In fact, these values result from having at least two changes (*i.e.,* four posts or comments) from different conditions.

## Changes in users' toxicity over time

Our collection's large volume of temporal data allows us to investigate toxicity changes over time. Therefore, we chose yearly intervals to answer RQ3 (*Does the toxicity of users change over time across different communities?*) and note observed changes in toxicity by computing the difference in the toxicity of every user across all the subreddits.

With Eq. (2), we computed users' toxicity percentages across all subreddits. Then, we calculated the change in toxicity ($\Delta$) in every pair of years for posting and commenting users. Subsequently, we used scatter plots to visualize the change in toxicity per user across subreddits, which we then converted to heatmaps with varying smoothing parameters. The heatmap plots in Fig. 5 show the distribution of toxicity in the posts and comments with smoothing at 64 neighbors. Figures 5A and 5B show the heatmap plots for the posts from the years 2007–2008 and 2018–2019, respectively, while Figs. 5C and 5D show the heatmap plots for the comments collected from the years 2007–2008 and 2018–2019. To clarify the

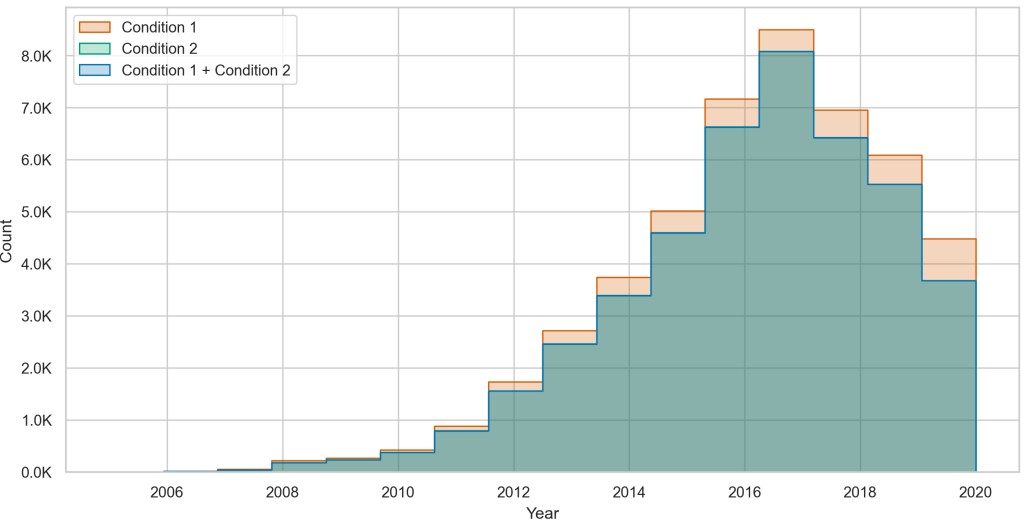

**Figure 4** The count of toxicity changes over time in commenting users from condition 1 (NT → T), condition 2 (T → NT), and both conditions combined.

observations, we removed users who showed no change in their toxicity in the plots (*i.e.,* users with $\Delta = 0$). Due to space limitations, we only show the heatmap plots from two pairs of years representing the collection's beginning and ending periods. The overall temporal analysis of the content shows that over time, changes in the toxicity of users' posts disperse across the participating communities, as illustrated by the increase in dark color in Figs. 5B and 5D from Fig. 5.

In other words, over time, more users diffuse their toxic behavior to a large number of varying communities. To further support this finding with users who leave comments, we conducted a dependent *T*-test on the commenting users' deltas for 2007 and 2019 (*i.e.,* their initial delta and final delta). Results show that at $p < 0.001$, the t-statistic value of the users is 57.031, indicating a significant change in the posting behavior of users, which any of the conditions mentioned earlier can describe (change from toxic to non-toxic or change from non-toxic to toxic within the same subreddit).

Lastly, we visualize the $\Delta$ values of posting users in Fig. 6, where we show the total change per year and interpolate the change by computing the smoothed rolling average on intervals of three years. The average line shows that changes peak in 2017 but drop after this point, suggesting some form of stability in the behavior of posting users. Similarly, Fig. 7 shows that the average toxicity change in commenting users peaks in 2018 and drops slightly after this year. Unlike posting users, commenting users continue to show high amounts of change despite the drop after 2018.

## Changes in toxicity and links

Originally, Reddit was a news-sharing website where users posted links to news websites or various multimedia content to instigate discussions with other users. Hence, most of Reddit's earlier content (primarily posts) contained links. However, links are not limited to

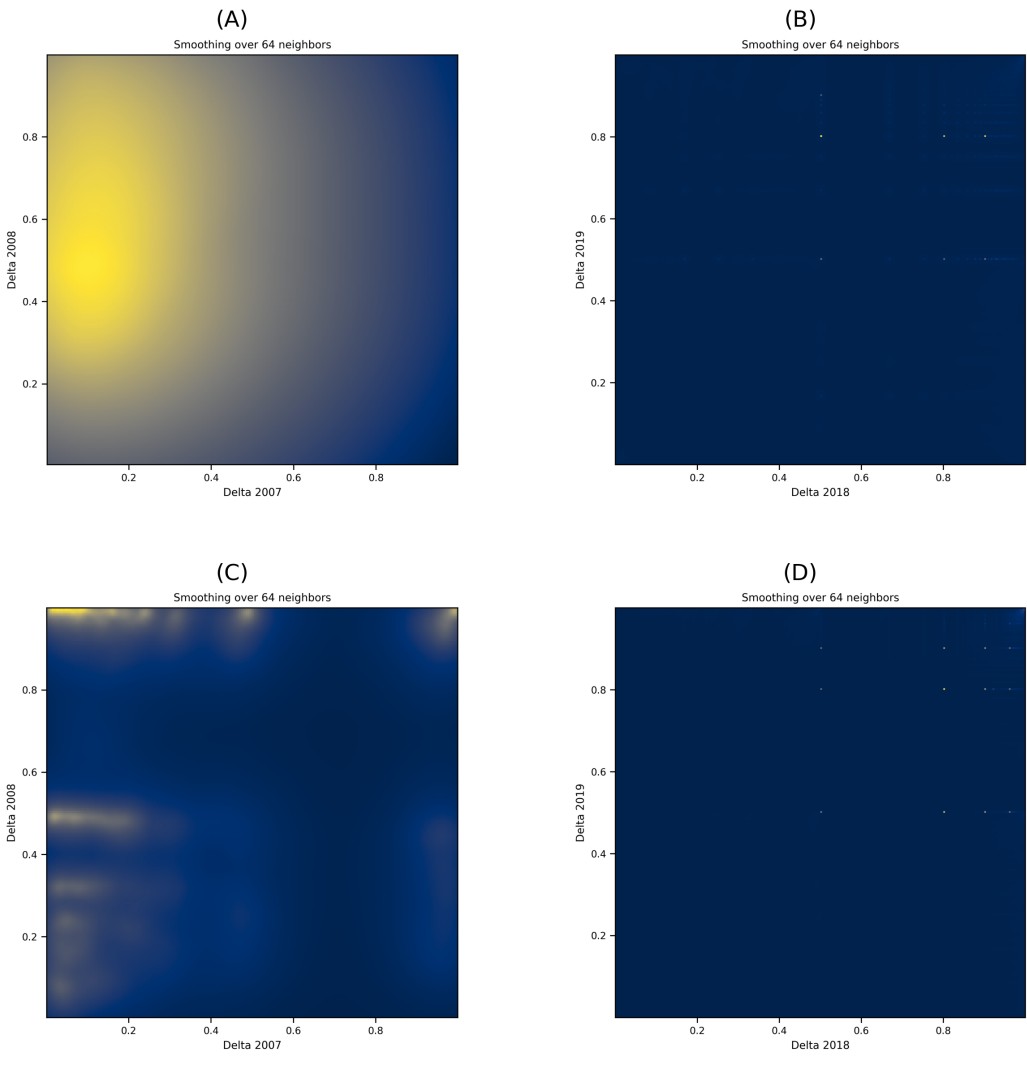

**Figure 5** **(A–D) Heatmap plots of the Δ in user posts and comments over two pairs of years.** The dark color in the heatmap plot denotes scattered deltas while the light colors denotes concentrated deltas in specific locations.

posts, as users can also include different types of links in their comments. Since our earlier investigation of toxic behavior focused on the textual content of user content, it is only natural to examine links to identify any correlation between toxicity in user content and certain types of links. To begin investigating links, we performed a preliminary exploratory analysis on the entire collection to identify the number of links and the percentage of links from the total number of posts and comments per year. The statistics illustrated in the top portion of Fig. 8 show the total number of posts, toxic posts, and links in each year, while the bottom portion of the figure shows the corresponding normalized (*i.e.,* scaled) totals using the minimum and maximum values from the totals. The accompanying values from Fig. 8 are in Appendix C, where we also show the percentage of toxic posts in Table C1. Statistics from Table C1 show that between 2005 and 2012, more than 50% of user posts

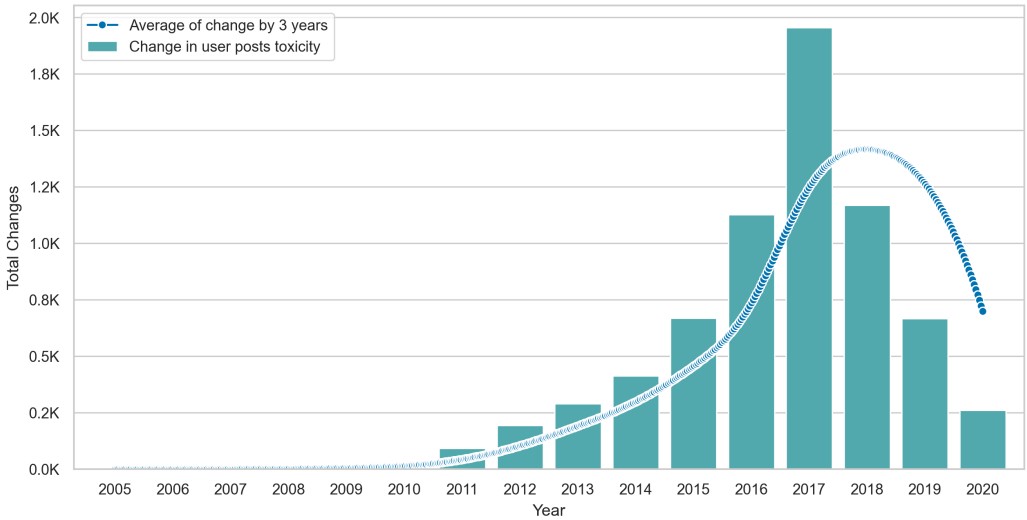

**Figure 6** The total amount of Δ in posting users content over time with an interpolation of Δ averages across three year intervals.

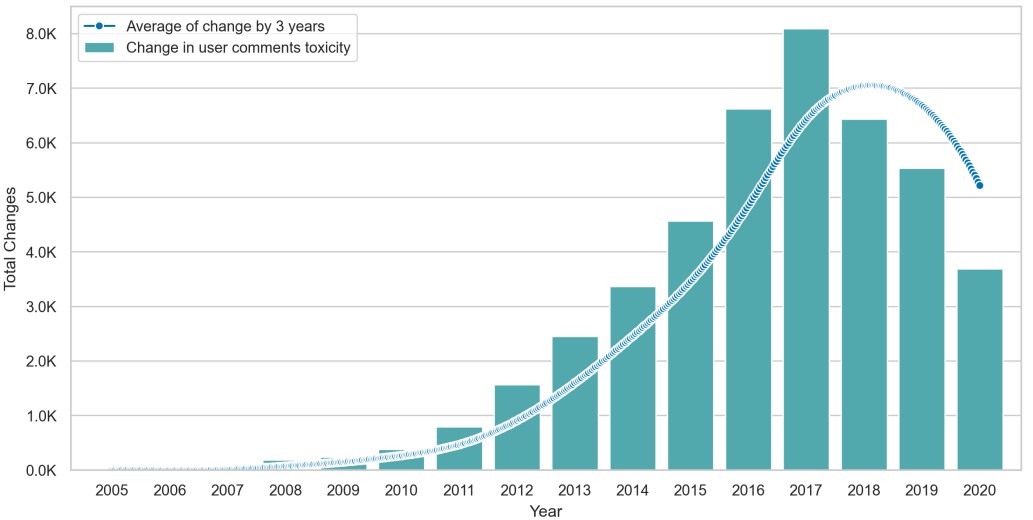

**Figure 7** The total amount of Δ in commenting users content over time with an interpolation of Δ averages across three year intervals.

contained links, which means that, indeed, posts from the earlier years of Reddit contained a significant amount of links. Upon further investigation of the links in user content, we found external links that redirect to websites outside Reddit and internal links that redirect to a Reddit user, post, community, or multimedia content uploaded on Reddit servers. Additionally, we found that some of the links in posts refer to videos, images, or other types of identifiable media.

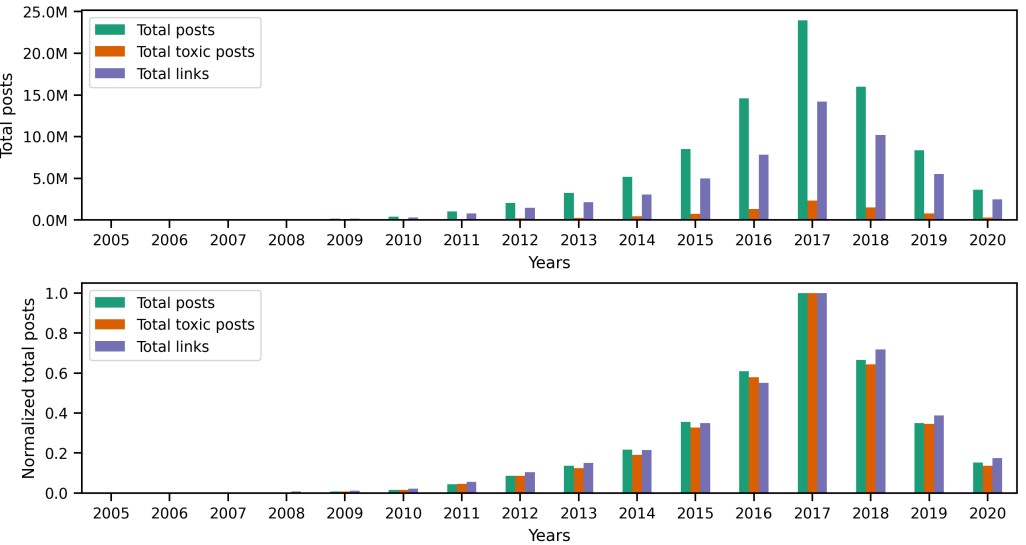

**Figure 8** The total number of posts, toxic posts, and links in every year followed by the normalized totals using the min-max scale.

Our findings from the posts collection in Fig. 9 and Table C2 show that around 84% of the links in posts are external. As for the remaining internal links, we found that in posts, they typically link to an image uploaded to the i.redd.it domain, or a video uploaded to the v.redd.it domain, or a Reddit user. Figure 9 also shows the total number of links with identifiable media types. We used the mimetypes python module to guess the media types from the link's text representation (*i.e.,* path). So, if a link address ends with .mp4, mimetypes identify it as a video without examining the link's content. In posts, we found that the most identifiable media type happens to be images, so we calculated the total number of links with images and the percentage of images from the known media types in links. The results in Table C2 show that most of the links in posts from later years contain images.

As for the comments collection, the statistics in Fig. 10 and Table C3 show that the percentage of links in comments is significantly less than that of links in posts. However, this outcome does not diminish the fact that more than 110 million comments contain links, which is about 5% of the entire comments collection. Furthermore, just like the posts collection, Fig. 11 shows that comments from earlier years in the collection contain more links than comments in later years.

The results in Table C4 show that around 90% of the links in comments are external, and out of all the media types identifiable in these links, images seem to appear the most in user comments. However, when comparing the percentage of images in posts and comments, around 83% of links in posts contain images, while 72% of links in comments contain images. This observation makes sense because many communities on Reddit, such as r/cringepics, require users to post images in the community.

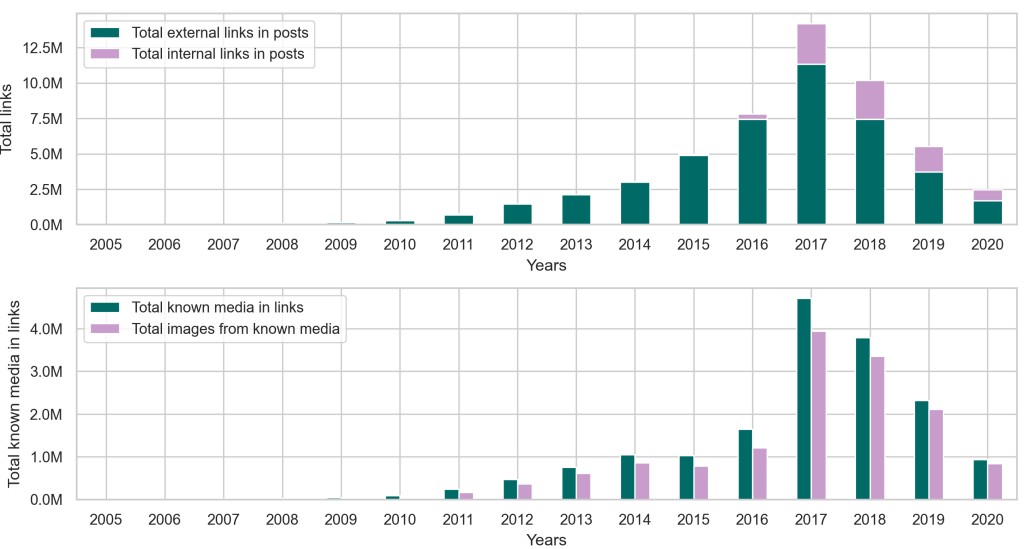

**Figure 9** The distribution of internal and external links, followed by the total number of known media types and image links from the posts collection.

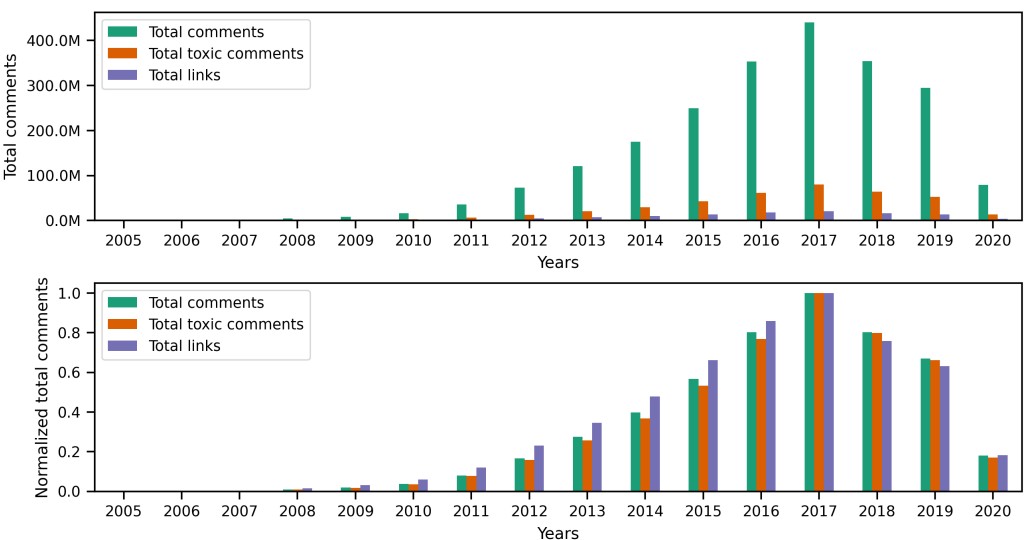

**Figure 10** The total number of comments, toxic comments, and links in every year followed by the normalized totals using the min-max scale.

After performing the preliminary exploratory analysis of links in the collection, we used the Granger causality test to find correlations between toxic behavior and links in posts and comments. First, we conducted a test between the volume of content (X) in each collection and the volume of toxic (both highly and slightly toxic) content (Y). Then, we conducted another test between the volume of links in each collection (Z) and the volume of toxic (both highly and slightly toxic) content (Y). While our original intention was to perform

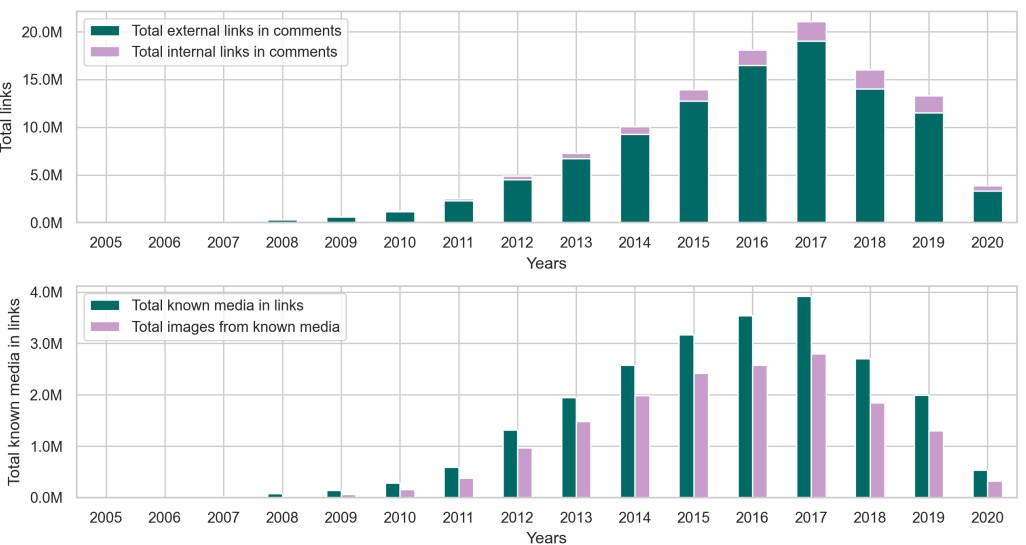

**Figure 11** The distribution of internal and external links, followed by the total number of known media types and image links from the comments collection.

the Granger causality test on user posts as well, we found that, since almost 99% of posts contain links, running the test will not provide us with valuable insights on the relationship between links in posts and toxicity. Moreover, our yearly time series in user posts did not produce a stationary series, which does not satisfy the requirement to conduct the Granger causality test. Therefore, we limit our experiments to user comments. Table 9 shows the F-statistic, $p$-value, and selected minimum lags in years from conducting the Granger causality test on the volumes of comments, toxic comments, and links in comments. The causality of toxicity in user comments can be observed in Table 9, where the $p$-valu $e < 0.05$ for the volume of comments and links. In other words, the volume of comments and links in comments influences the volume of toxic comments in the collection.

## IMPLICATIONS AND CONCLUSIONS

In this research, using over 10 thousand labeled comments, we trained feature-based and neural network-based models to infer the toxicity of user posts. We then used a fine-tuned BERT model to analyze a large-scale Reddit collection of more than two billion posts from more than 1.2 million users over 16 years for a detailed analysis of the toxic behavior of users in multiple online communities. Our contributions are three-fold:

- First, to our knowledge, we built one of the biggest labeled datasets of Reddit toxic comments and made it publicly available for further research (https://github.com/Hind-Almerekhi/toxicityChangesReddit). Additionally, compared to other binary labeled datasets, our dataset contains three levels of toxicity for each comment, ranging from non-toxic to slightly toxic to highly toxic.

**Table 9  Results of Granger causality for the comments collection at a minimum lag in years.**

|  | F-statistic | $p$-value | Lags (years) |
|---|---|---|---|
| (X,Y): X $\rightarrow$ Y | 7.6306 | 0.014 | 2 |
| (Z,Y): Z $\rightarrow$ Y | 10.3849 | 0.014 | 3 |

- Second, by systematic comparisons of common feature-based models and neural network-based models, we demonstrate that a fine-tuned BERT model performs best for toxicity detection in posts and comments from Reddit.
- Third, our work is one of the first large-scale studies that investigate toxic behavior across multiple communities. We start with a list of cross-community users from the top 100 subreddits and expand our collection by obtaining posts and comments from more than 107,000 subreddits to reveal how users behave across communities from the perspective of toxicity.

## Implications

Our work has several implications for the safety and moderation of online communities. These implications include the following:

### *Early detection of changes in toxicity*

The dissemination of toxicity in online communities impacts the positive experience many users seek when using social media platforms. Several research studies showed that users could negatively influence each other when interacting in online communities (*Kwon & Gruzd, 2017*; *Zhang et al., 2018*; *Cheng et al., 2017*).

This type of negative behavior can continue to spread and harm online communities. Monitoring the change in users' toxicity can be an early detection method for toxicity in online communities. The proposed methodology can identify when users exhibit a change by calculating the toxicity percentage in posts and comments. This change, combined with the toxicity level our system detects in users' posts, can be used efficiently to stop toxicity dissemination. Furthermore, our methodology supports detecting toxicity early in online communities from users' toxicity. In an active setting, users' toxicity percentages can issue early alerts to online community moderators (bots or humans) so they can investigate potential toxicity incidents and take necessary actions to mitigate the further spread of toxicity in communities.

### *Aid moderators with toxicity changes*

Judging the toxicity of user content may not always be ideal for preventing the spread of hate and incivility (*Rajadesingan, Resnick & Budak, 2020*). Our study showed that users changed their posts' toxicity within and across different communities. This change can result from fluctuations in the users' feelings or changes in the atmosphere of their communities. This change, coupled with the toxicity of the users' content, can create an accurate assessment method to prevent the spread of toxicity. For instance, instead of banning a user for a tasteless contribution they left once, moderators can consider the users' predominant toxicity and that of their previous content. This approach will

(A)

(B)

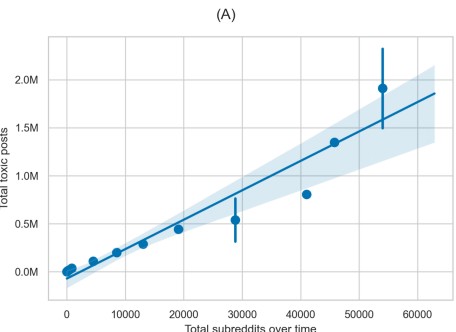
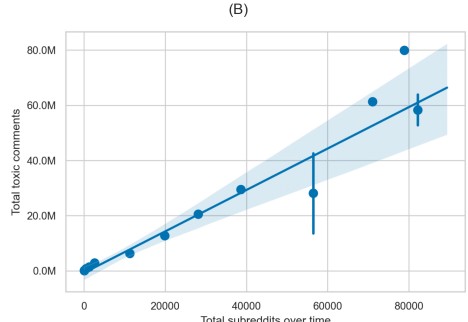

**Figure 12** Correlation between the total number of participating subreddits over time and (A) the total number of toxic posts and (B) the total number of toxic comments.

prevent automated bots and moderators from excessively penalizing or banning users. This sophisticated user- and content-based toxicity assessment allows moderators to control toxicity and detect malicious users who deserve banning from online communities. Trolls and the like (*Cheng et al., 2017*) can also be prevented from polluting online communities by using our recommended method to judge users based on their content's predominant toxicity.

Moreover, the rules and norms of communities can be changed to prevent the spread of toxicity (*Chandrasekharan et al., 2018*). For example, when users show a rapid change in the toxicity of their behavior, the moderation ecosystem might raise alerts/reminders for any breaches that do not conform to the community norms. Lastly, our study suggests that one way to limit the spread of toxicity is by limiting the spaces (*i.e.,* communities) in which users can participate. To illustrate this finding, in Fig. 12 we show the correlation between the total amount of toxic posts (Fig. 12A), comments (Fig. 12B) and the total number of communities that users participate in over time. The figures show a positive correlation between the increase in the number of communities and the increase in toxicity. Ultimately, we cannot guarantee that this is the only reason behind the increase in toxic content, yet we argue that increasing communities could allow users to spread toxic content.

## Limitations and future work

Since our research focuses mainly on text analysis to detect toxicity, one limitation is that toxicity takes different forms (*e.g.,* images, videos, and sound clips). While more sophisticated techniques are required to examine and analyze such content (*Rafiq et al., 2015*), multimedia submissions also have text titles that we studied in this work. Another limitation of this work is that it does not fully consider bias in the toxicity of the labels we obtained through crowdsourcing (*Vaidya, Mai & Ning, 2020*; *Hanu, Thewlis & Haco, 2021*). However, since toxicity is a subjective matter, our study performed toxicity detection in a simplified manner without accounting for subjectivity (*Zhao, Zhang & Hopfgartner, 2022*). Lastly, we note that our study did not tackle any contextual or categorical characteristics of toxic content (*Radfar, Shivaram & Culotta, 2020*). That is partially due to the heterogeneous nature of most Reddit communities, making it extremely

difficult to capture their context to judge different types of content, such as profanity in certain Not Safe For Work (NSFW) communities (*Madukwe & Gao, 2019*).

Upon investigating the toxic posting behavior of users, we came across several ideas that can lead to interesting future research directions. One of the ideas focuses on different scenarios involving users joining new communities and considering the changes in their toxicity to these new communities (*Cheng, Danescu-Niculescu-Mizil & Leskovec, 2015*; *Choi et al., 2015*; *Cheng et al., 2017*). Another take on the problem of toxic posting behavior can focus on specific topics within each community (*e.g.*, controversial topics or hot news) to study how they trigger toxicity within users, as opposed to noncontroversial or regular topics (*e.g.*, entertainment news or funny stories). Besides focusing on various topics within online communities, one can also study the temporal characteristics that foster the evolution of toxic communities from a few users with predominantly toxic posts. Considering the factors that contribute to the rapid growth of such toxic communities is also necessary for providing moderators and platform designers with the right tools to prevent toxicity from contaminating online communities.

## CONCLUSIONS

In this research, we investigated users' toxic cross-community behavior based on the toxicity of their posts and comments. Our fine-tuned BERT model achieved a classification accuracy of 91.27% and an average $F_1$ score of 0.79, showing a 2% and 7% improvement in performance compared to the best-performing baseline models based on neural networks. We addressed RQ1 by running a prediction experiment on the posts and comments from our Reddit collection. The analysis showed that 9.33% of the posts are toxic, and 17.6% of the comments are toxic. We answered RQ2 by investigating the changes in the toxicity of users' content across communities based on two primary conditions. First, our analysis showed that 30.68% of posting users showed changes in their toxicity levels. Moreover, 81.67% of commenting users showed changes in their toxicity levels, mainly across multiple communities. Moreover, we found through answering RQ3 that, over time, toxicity disperses with an increase in the number of participating users and the frequency of cross-community participation. This finding is helpful because it can provide community moderators with leads to help them track patterns from active users to prevent them from spreading toxic content online.

Lastly, we conducted a Granger causality test between the volume of comments, the volume of links in comments, and the volume of toxicity. We found that links in comments can influence toxicity within those comments. This research addresses a prominent issue in social media platforms: toxic behavior negatively impacts other users' experience. Thus, we believe it is necessary to conduct more research on users' toxic behavior to help us understand the behavior's dynamics.

## APPENDIX A

**Table A1** The top 100 (1–50) subreddits ranked by the total number of subscribers along with the total number of posts and comments from our dataset.

| Rank | Subreddit | Subscribers | Posts | Comments |
|------|-----------|-------------|-------|----------|
| 1 | r/funny | 17,934,343 | 1,226,923 | 38,340,865 |
| 2 | r/AskReddit | 17,829,339 | 3,571,863 | 176,219,659 |
| 3 | r/todayilearned | 17,658,854 | 379,675 | 24,982,380 |
| 4 | r/science | 17,599,931 | 133,623 | 4,387,503 |
| 5 | r/worldnews | 17,522,963 | 613,496 | 31,844,773 |
| 6 | r/pics | 17,517,767 | 905,005 | 35,795,690 |
| 7 | r/IAmA | 17,215,966 | 34,839 | 7,650,945 |
| 8 | r/gaming | 16,860,176 | 762,776 | 25,502,647 |
| 9 | r/videos | 16,758,252 | 838,316 | 23,327,414 |
| 10 | r/movies | 16,468,377 | 407,215 | 17,824,942 |
| 11 | r/aww | 15,907,509 | 629,697 | 8,902,569 |
| 12 | r/Music | 15,879,988 | 400,517 | 4,567,182 |
| 13 | r/gifs | 15,051,932 | 175,824 | 12,300,726 |
| 14 | r/news | 14,994,220 | 896,236 | 25,720,157 |
| 15 | r/explainlikeimfive | 14,671,688 | 343,303 | 6,037,936 |
| 16 | r/askscience | 14,587,860 | 253,372 | 1,609,707 |
| 17 | r/EarthPorn | 14,197,666 | 92,811 | 1,178,788 |
| 18 | r/books | 13,699,914 | 83,049 | 2,786,204 |
| 19 | r/television | 13,617,822 | 129,974 | 6,005,034 |
| 20 | r/LifeProTips | 13,212,746 | 113,033 | 3,541,704 |
| 21 | r/mildlyinteresting | 13,170,674 | 319,910 | 7,369,887 |
| 22 | r/space | 12,915,830 | 77,426 | 2,126,598 |
| 23 | r/Showerthoughts | 12,797,856 | 944,483 | 9,315,365 |
| 24 | r/DIY | 12,753,699 | 52,804 | 1,647,926 |
| 25 | r/Jokes | 12,604,471 | 290,732 | 2,985,167 |
| 26 | r/sports | 12,552,016 | 129,084 | 2,422,532 |
| 27 | r/gadgets | 12,518,386 | 43,995 | 1,420,475 |
| 28 | r/tifu | 12,504,386 | 53,468 | 3,271,900 |
| 29 | r/nottheonion | 12,451,389 | 112,276 | 4,024,654 |
| 30 | r/InternetIsBeautiful | 12,433,228 | 29,919 | 338,448 |
| 31 | r/photoshopbattles | 12,363,401 | 93,001 | 624,593 |
| 32 | r/history | 12,356,415 | 48,904 | 1,105,441 |
| 33 | r/food | 12,351,889 | 181,129 | 2,401,415 |
| 34 | r/Futurology | 12,332,702 | 88,915 | 3,437,276 |
| 35 | r/Documentaries | 12,298,293 | 47,042 | 1,539,521 |
| 36 | r/dataisbeautiful | 12,293,355 | 36,084 | 2,016,558 |
| 37 | r/listentothis | 12,243,720 | 143,301 | 379,788 |
| 38 | r/UpliftingNews | 12,213,060 | 56,064 | 1,495,984 |
| 39 | r/personalfinance | 12,212,893 | 139,922 | 4,497,818 |

**Table A1** (*continued*)

| Rank | Subreddit | Subscribers | Posts | Comments |
|------|-----------|-------------|-------|----------|
| 40 | r/GetMotivated | 12,135,696 | 50,356 | 935,431 |
| 41 | r/OldSchoolCool | 12,083,017 | 107,293 | 3,061,358 |
| 42 | r/philosophy | 12,081,626 | 23,948 | 713,078 |
| 43 | r/Art | 11,868,112 | 151,648 | 926,225 |
| 44 | r/nosleep | 11,678,653 | 23,966 | 607,853 |
| 45 | r/creepy | 11,665,125 | 46,469 | 1,147,867 |
| 46 | r/WritingPrompts | 11,612,067 | 206,881 | 948,065 |
| 47 | r/TwoXChromosomes | 11,215,698 | 52,283 | 3,002,720 |
| 48 | r/Fitness | 6,186,196 | 115,007 | 4,672,605 |
| 49 | r/technology | 5,551,587 | 267,264 | 9,225,651 |
| 50 | r/WTF | 4,861,274 | 239,730 | 19,032,277 |

**Table A2** The top 100 (51–100) subreddits ranked by the total number of subscribers along with the total number of posts and comments from our dataset.

| Rank | Subreddit | Subscribers | Submissions | Comments |
|---|---|---|---|---|
| 51 | r/bestof | 4,772,718 | 41,570 | 1,869,232 |
| 52 | r/AdviceAnimals | 4,322,195 | 491,179 | 17,734,082 |
| 53 | r/politics | 3,468,561 | 955,613 | 60,494,833 |
| 54 | r/atheism | 2,096,408 | 147,972 | 8,890,785 |
| 55 | r/europe | 1,526,462 | 123,920 | 5,707,891 |
| 56 | r/interestingasfuck | 1,385,740 | 73,778 | 3,595,720 |
| 57 | r/woahdude | 1,345,016 | 58,643 | 1,514,561 |
| 58 | r/leagueoflegends | 1,118,408 | 550,318 | 22,450,314 |
| 59 | r/gameofthrones | 1,116,208 | 110,096 | 4,023,561 |
| 60 | r/pcmasterrace | 1,103,955 | 392,241 | 11,081,807 |
| 61 | r/BlackPeopleTwitter | 1,073,938 | 67,891 | 5,072,940 |
| 62 | r/reactiongifs | 1,038,629 | 86,617 | 1,488,555 |
| 63 | r/trees | 1,006,481 | 287,696 | 6,251,863 |
| 64 | r/Unexpected | 965,760 | 45,269 | 1,404,524 |
| 65 | r/Overwatch | 948,162 | 329,076 | 6,781,358 |
| 66 | r/oddlysatisfying | 905,675 | 53,992 | 1,683,927 |
| 67 | r/Android | 897,620 | 113,772 | 6,668,499 |
| 68 | r/wholesomememes | 840,077 | 36,283 | 1,003,541 |
| 69 | r/Games | 839,529 | 158,645 | 8,505,029 |
| 70 | r/programming | 826,809 | 51,209 | 3,208,560 |
| 71 | r/4chan | 819,656 | 38,445 | 2,338,530 |
| 72 | r/nba | 805,171 | 295,232 | 28,721,576 |
| 73 | r/facepalm | 791,286 | 36,112 | 1,955,898 |
| 74 | r/cringepics | 780,791 | 30,813 | 2,132,087 |
| 75 | r/me_irl | 779,311 | 435,999 | 2,081,621 |
| 76 | r/relationships | 774,812 | 61,107 | 5,975,894 |
| 77 | r/sex | 761,247 | 39,179 | 2,305,426 |
| 78 | r/pokemon | 760,949 | 120,266 | 3,992,989 |
| 79 | r/fffffffuuuuuuuuuuuu | 759,747 | 53,360 | 2,372,289 |
| 80 | r/lifehacks | 755,376 | 9,845 | 480,229 |
| 81 | r/Frugal | 741,976 | 24,157 | 1,611,244 |
| 82 | r/soccer | 736,005 | 231,161 | 20,685,778 |
| 83 | r/tattoos | 732,943 | 29,924 | 470,543 |
| 84 | r/pokemongo | 730,140 | 115,926 | 2,434,360 |
| 85 | r/comics | 726,976 | 85,320 | 1,128,707 |
| 86 | r/OutOfTheLoop | 688,156 | 55,979 | 1,249,050 |
| 87 | r/malefashionadvice | 684,010 | 56,821 | 2,314,703 |
| 88 | r/CrappyDesign | 667,846 | 78,494 | 1,376,476 |
| 89 | r/StarWars | 658,622 | 121,300 | 3,685,999 |
| 90 | r/YouShouldKnow | 644,359 | 8,068 | 577,679 |

**Table A2** (*continued*)

| Rank | Subreddit | Subscribers | Submissions | Comments |
|------|-----------|-------------|-------------|----------|
| 91 | r/AskHistorians | 637,383 | 106,763 | 629,207 |
| 92 | r/buildapc | 635,055 | 281,713 | 5,049,608 |
| 93 | r/nfl | 626,637 | 197,693 | 33,747,257 |
| 94 | r/HistoryPorn | 626,507 | 33,838 | 647,963 |
| 95 | r/RoastMe | 622,922 | 23,436 | 1,808,683 |
| 96 | r/loseit | 613,079 | 47,978 | 1,388,423 |
| 97 | r/FoodPorn | 612,361 | 39,742 | 570,383 |
| 98 | r/AnimalsBeingJerks | 605,103 | 11,596 | 429,629 |
| 99 | r/dankmemes | 598,376 | 238,182 | 2,003,893 |
| 100 | r/rickandmorty | 586,805 | 51,538 | 1,017,558 |

# APPENDIX B

## Is This Askreddit Comment Toxic? If So, How Toxic Is It?

Instructions ▲

**Overview**

In this task, you will be asked to label /r/AskReddit comments as either toxic or not toxic. In this context, a comment is considered toxic if it was **"a rude, disrespectful, or unreasonable comment that is likely to make you leave a discussion."** (definition taken from Google's Perspective API). If a comment was toxic, you will be asked to rate how toxic the comment is according to the steps given below.

**Steps**

1. Read the AskReddit comments carefully before deciding if they are toxic or not toxic.
2. To understand the context of the comment, you **should** click on the comment's URL to view the AskReddit discussion thread.
3. Select one option only: a comment can be either toxic or not toxic. If it is toxic, rate it's toxicity.
4. If a comment was toxic, it can be slightly toxic (rated as 1) or highly toxic (rated as 2).

**Rules and Tips**

**Do This**

Read the full comment before deciding if it is toxic or not. Always refer to the definition of toxic comments (given in the overview section) if you get stuck or confused. If the comment was ambiguous, click on the comment to jump to the conversation thread; read previous comments along with the AskReddit thread title to understand the context.

**Be Careful Of**

Avoiding to read the comment within context will result in failing the test. Please make sure you click on the comment and view the conversation thread to ensure you don't make mistakes while labeling.

**Do Not**

Randomly click on labels like toxic or not toxic. This will result in you failing the test. You are forced to click on all the comments and spend some time to read and label accordingly.

**Examples**

In each example, try viewing the comment within context by clicking on the comment's text (i.e. URL), which will redirect you to the original AskReddit thread in which the comment was posted.

Example 1

Agree. It drives me nuts when someone is driving with considerably less than 200km/h on the left lane of the only part of Autobahn I use that has no speed limit.
The comment above is **not toxic** because it is not rude or offensive to the reader.

Example 2

Drones. I have race quads and a lot of us practice safety and follow the rules. But there are alot of idiots who take their photography rigs and fly over crowded areas or close to people because they think it will make for cool footage. It's really dangerous and their carelessness effects everyone in the drone community the same. More regulation and it puts the responsible flyers in the negative light as well. These things can cause serious injuries. The ones we fly aren't toys. But that doesn't matter to the idiots who want that awesome footage.
The comment above is **toxic** because it is somewhat rude and offensive to the reader. As for the rating of the comment, it is considered **slightly toxic** because most readers can tolerate the toxic language present in the comment.

Example 3

Saw Thor Ragnarok on monday in the middle of the day, though I gambled and lost as I didn't realise it was still half term for some kids around here. Kid in there would not fucking shut up. Hela comes on screen- "Mum, is she evil?" and other such inane questions. Shut. The. Fuck. Up.
The comment above is **toxic** because it is rude and offensive to the reader. As for the rating of the comment, it is considered **highly toxic** because most readers cannot tolerate the toxic language present in the comment.

**Figure B1** **The labeling task instructions that we provided to crowd workers.**

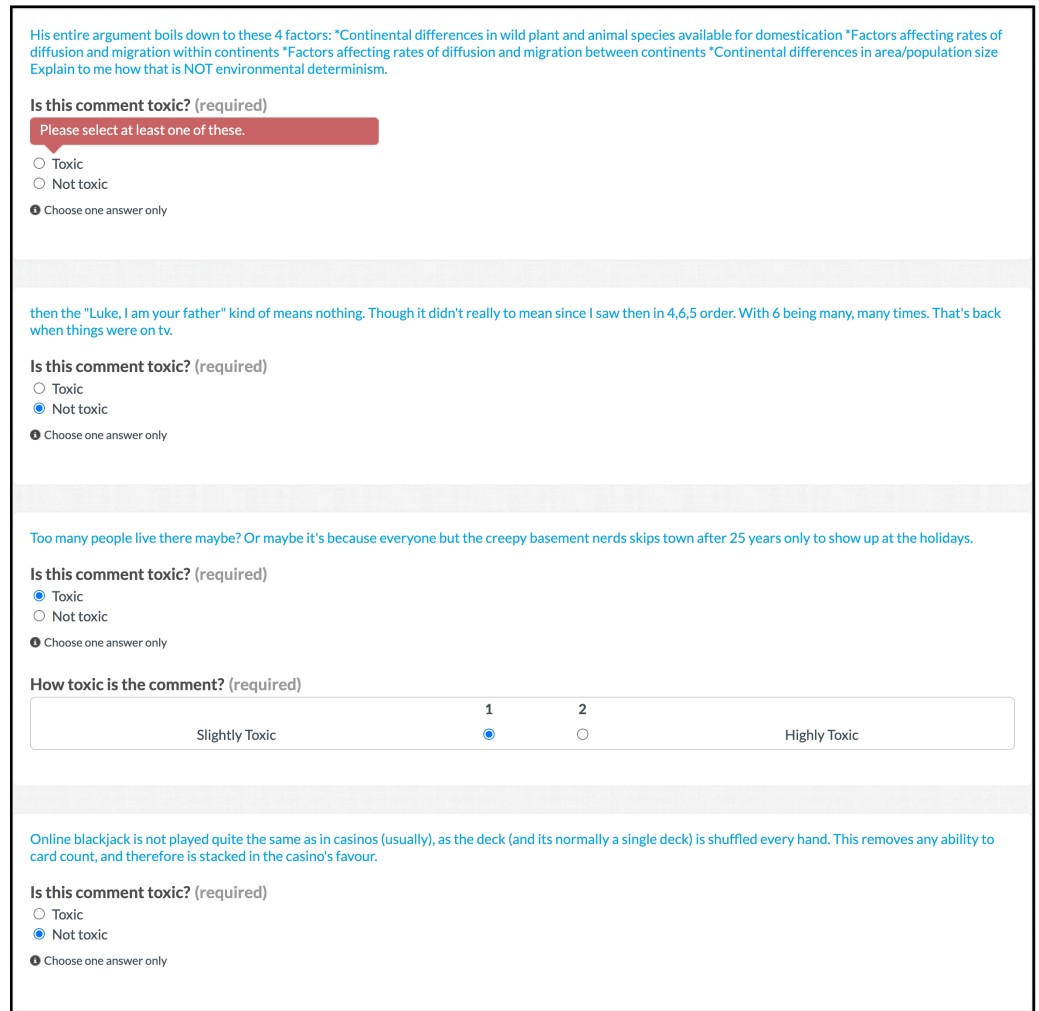

**Figure B2** **The validation questions that crowd workers had to pass before beginning labeling.**

# APPENDIX C

**Table C1 The total number of posts, toxic posts, and links in every year followed by the normalized totals using the min-max scale.**

| Year | Original counts | | | Normalized counts | | |
|---|---|---|---|---|---|---|
| | Posts | Toxic (%) | Links (%) | Posts | Toxic | Links |
| 2005 | 1,690 | 81 (4.79) | 1,690 (100) | 0.0 | 0.0 | 0.0 |
| 2006 | 10,917 | 895 (8.2) | 10,917 (100) | 0.0004 | 0.0004 | 0.0006 |
| 2007 | 47,556 | 4,372 (9.19) | 47,556 (100) | 0.0019 | 0.0018 | 0.0032 |
| 2008 | 105,825 | 10,407 (9.83) | 101,357 (95.78) | 0.0043 | 0.0044 | 0.007 |
| 2009 | 188,171 | 18,344 (9.75) | 167,255 (88.88) | 0.0078 | 0.0079 | 0.0117 |
| 2010 | 385,611 | 37,612 (9.75) | 305,616 (79.26) | 0.016 | 0.0162 | 0.0214 |
| 2011 | 1,058,576 | 108,253 (10.23) | 791,010 (74.72) | 0.0441 | 0.0466 | 0.0556 |
| 2012 | 2,052,406 | 201,278 (9.81) | 1,472,632 (71.75) | 0.0856 | 0.0866 | 0.1036 |
| 2013 | 3,247,906 | 287,250 (8.84) | 2,133,632 (65.69) | 0.1355 | 0.1236 | 0.1501 |
| 2014 | 5,176,179 | 442,283 (8.54) | 3,054,626 (59.01) | 0.2159 | 0.1904 | 0.215 |
| 2015 | 8,532,341 | 762,746 (8.94) | 4,978,694 (58.35) | 0.3559 | 0.3284 | 0.3505 |
| 2016 | 14,613,378 | 1,346,608 (9.21) | 7,822,994 (53.53) | 0.6097 | 0.5797 | 0.5508 |
| 2017 | 23,967,825 | 2,322,698 (9.69) | 14,201,243 (59.25) | 1.0 | 1.0 | 1.0 |
| 2018 | 15,968,062 | 1,496,033 (9.37) | 10,201,097 (63.88) | 0.6662 | 0.6441 | 0.7183 |
| 2019 | 8,379,712 | 804,135 (9.6) | 5,523,004 (65.91) | 0.3496 | 0.3462 | 0.3888 |
| 2020 | 3,640,757 | 315,212 (8.66) | 2,468,515 (67.8) | 0.1518 | 0.1357 | 0.1737 |

**Table C2 The total number of internal links, external links, known media type links, and image links from the posts collection.**

| Year | Categories of links | | Contents of links | |
|---|---|---|---|---|
| | Internal | External | Known media (%) | Images (%) |
| 2005 | 0 | 1,690 | 565 (33.43) | 9 (1.59) |
| 2006 | 0 | 10,917 | 3,392 (31.07) | 149 (4.39) |
| 2007 | 0 | 47,556 | 13,001 (27.34) | 1,267 (9.75) |
| 2008 | 57 | 101,300 | 28,890 (28.5) | 3,272 (11.33) |
| 2009 | 197 | 167,058 | 45,240 (27.05) | 8,270 (18.28) |
| 2010 | 282 | 305,334 | 93,788 (30.69) | 38,803 (41.37) |
| 2011 | 81,426 | 709,584 | 245,682 (31.06) | 166,934 (67.95) |
| 2012 | 550 | 1,472,082 | 471,553 (32.02) | 364,726 (77.35) |
| 2013 | 1,078 | 2,132,554 | 751,834 (35.24) | 610,537 (81.21) |
| 2014 | 30,615 | 3,024,011 | 1,052,430 (34.45) | 856,061 (81.34) |
| 2015 | 70,917 | 4,907,777 | 1,032,840 (20.75) | 788,774 (76.37) |
| 2016 | 389,853 | 7,433,141 | 1,643,321 (21.01) | 1,211,723 (73.74) |
| 2017 | 2,862,721 | 11,338,522 | 4,710,886 (33.17) | 3,940,218 (83.64) |
| 2018 | 2,757,591 | 7,443,506 | 3,788,301 (37.14) | 3,354,431 (88.55) |
| 2019 | 1,803,981 | 3,719,023 | 2,323,104 (42.06) | 2,109,346 (90.8) |
| 2020 | 771,499 | 1,697,016 | 934,850 (37.87) | 842,602 (90.13) |

**Table C3** The total number of comments, toxic comments, and links in every year followed by the normalized totals using the min-max scale.

| | Original counts | | | Normalized counts | | |
|---|---|---|---|---|---|---|
| Year | Comments | Toxic (%) | Links (%) | Comments | Toxic | Links |
| 2005 | 310 | 26 (8.39) | 38 (12.26) | 0.0 | 0.0 | 0.0 |
| 2006 | 169,608 | 17,553 (10.35) | 19,052 (11.23) | 0.0004 | 0.0002 | 0.0009 |
| 2007 | 849,828 | 119,477 (14.06) | 74,055 (8.71) | 0.0019 | 0.0015 | 0.0035 |
| 2008 | 4,573,561 | 795,377 (17.39) | 338,229 (7.4) | 0.0104 | 0.01 | 0.0161 |
| 2009 | 8,494,022 | 1,450,220 (17.07) | 674,146 (7.94) | 0.0193 | 0.0181 | 0.032 |
| 2010 | 16,384,988 | 2,830,121 (17.27) | 1,269,001 (7.74) | 0.0372 | 0.0354 | 0.0602 |
| 2011 | 35,473,547 | 6,260,102 (17.65) | 2,531,916 (7.14) | 0.0806 | 0.0783 | 0.1202 |
| 2012 | 72,943,244 | 12,704,594 (17.42) | 4,883,585 (6.7) | 0.1657 | 0.1589 | 0.2318 |
| 2013 | 121,155,630 | 20,530,863 (16.95) | 7,262,543 (5.99) | 0.2752 | 0.2569 | 0.3447 |
| 2014 | 175,223,888 | 29,461,807 (16.81) | 10,063,833 (5.74) | 0.398 | 0.3686 | 0.4777 |
| 2015 | 249,496,457 | 42,597,414 (17.07) | 13,934,125 (5.58) | 0.5667 | 0.5329 | 0.6614 |
| 2016 | 352,996,950 | 61,346,847 (17.38) | 18,100,953 (5.13) | 0.8017 | 0.7675 | 0.8591 |
| 2017 | 440,297,137 | 79,930,465 (18.15) | 21,068,587 (4.79) | 1.0 | 1.0 | 1.0 |
| 2018 | 353,701,991 | 63,881,250 (18.06) | 15,995,772 (4.52) | 0.8033 | 0.7992 | 0.7592 |
| 2019 | 294,450,367 | 52,818,942 (17.94) | 13,309,224 (4.52) | 0.6688 | 0.6608 | 0.6317 |
| 2020 | 79,370,258 | 13,603,534 (17.14) | 3,867,698 (4.87) | 0.1803 | 0.1702 | 0.1836 |

**Table C4** The total number of internal links, external links, known media type links, and image links from the comments collection.

| | Categories of links | | Contents of links | |
|---|---|---|---|---|
| Year | Internal | External | Known media (%) | Images (%) |
| 2005 | 0 | 38 | 11 (28.95) | 0 (0.0) |
| 2006 | 1 | 19,051 | 4,679 (24.56) | 556 (11.88) |
| 2007 | 2 | 74,053 | 17,253 (23.3) | 3,420 (19.82) |
| 2008 | 12,299 | 325,930 | 74,095 (21.91) | 23,315 (31.47) |
| 2009 | 54,658 | 619,488 | 141,166 (20.94) | 60,645 (42.96) |
| 2010 | 99,215 | 1,169,786 | 280,332 (22.09) | 154,372 (55.07) |
| 2011 | 206,306 | 2,325,610 | 590,792 (23.33) | 378,633 (64.09) |
| 2012 | 370,074 | 4,513,511 | 1,317,284 (26.97) | 970,935 (73.71) |
| 2013 | 558,468 | 6,704,073 | 1,944,145 (26.77) | 1,479,230 (76.09) |
| 2014 | 779,987 | 9,283,846 | 2,575,111 (25.59) | 1,988,173 (77.21) |
| 2015 | 1,181,543 | 12,752,582 | 3,171,927 (22.76) | 2,416,944 (76.2) |
| 2016 | 1,630,462 | 16,470,491 | 3,538,101 (19.55) | 2,577,966 (72.86) |
| 2017 | 2,055,250 | 19,013,337 | 3,920,151 (18.61) | 2,801,227 (71.46) |
| 2018 | 1,974,185 | 14,021,587 | 2,702,087 (16.89) | 1,840,742 (68.12) |
| 2019 | 1,776,779 | 11,532,445 | 1,990,511 (14.96) | 1,300,974 (65.36) |
| 2020 | 533,875 | 3,333,823 | 536,813 (13.88) | 323,796 (60.32) |

### Funding

This publication was funded by Qatar Research Leadership program grant, ID# QRLP9-G-3330102, from the Qatar National Research Fund (a member of Qatar Foundation). The funders had no role in study design, data collection and analysis, decision to publish, or preparation of the manuscript.

### Grant Disclosures

The following grant information was disclosed by the authors:
Qatar Research Leadership program grant from the Qatar National Research Fund (a member of Qatar Foundation): QRLP9-G-3330102.

### Competing Interests

The authors declare there are no competing interests.

### Author Contributions

- Hind Almerekhi conceived and designed the experiments, performed the experiments, analyzed the data, performed the computation work, prepared figures and/or tables, authored or reviewed drafts of the article, and approved the final draft.
- Haewoon Kwak conceived and designed the experiments, analyzed the data, authored or reviewed drafts of the article, and approved the final draft.
- Bernard J. Jansen conceived and designed the experiments, analyzed the data, authored or reviewed drafts of the article, and approved the final draft.

### Data Availability

Data is available at Github:
https://github.com/Hind-Almerekhi/toxicityChangesReddit.

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
