# Peer review of "Investigating toxicity changes of cross-community redditors from 2 billion posts and comments"

_PeerJ Computer Science, doi:10.7717/peerj-cs.1059_

## Round 0.1 · original submission · Major Revisions

All reviewers value the work presented in this submission as being interesting, but they do have a number of concerns that would need to be addressed in a major revision.

These concerns are linked, among others, to the need for justification of how representative the Reddit sample used was, further detail / justification on the decisions made in the annotation process, the definition of posts used, as well as missing references. Please carefully consider reviewer comments in preparing a revision.

Reviewer 1 ·

Basic reporting

The authors' definition of "post" is inconsistent with Reddit's own. As stated on https://www.reddithelp.com/hc/en-us/sections/201015409-Posting-Commenting, there are two types of content on Reddit: "posts" (top-level submissions that can be either one of Post, Images & Videos, Link, or Poll) and "comments" (any responses by other users as well as the original poster to a post). Take this u/reddit profile page (https://www.reddit.com/user/reddit), which is one of the Reddit admin accounts, for example. The profile page will display "POSTS" and "COMMENTS" as two separate tabs, displaying the history of u/reddit's posts and comments according to their respective official definitions as I mentioned above. In other words, the authors' definition of "post" that includes both posts/submissions and comments is incorrect. On Reddit, submissions are posts, but comments are not.

Having said that, I strongly recommend that the authors modify their definitions to align with Reddit's official definition, including their use of the phrase "2 billion posts" in the title.

Experimental design

Analyzing the top 100 subreddits based on subscriber count can indeed provide interesting insights into what is happening in those large subreddits, but it can still not give a complete story in terms of the overall landscape of the entire Reddit. There are two main reasons why. 

First, as per https://frontpagemetrics.com/, there are more than 3.4 million subreddits, 138,000 of which are active. As an avid Redditor myself, I can attest that different subreddits can have very different nuance in their posts and comments. In terms of their active users, some subreddits are quite heterogenous (e.g., r/news, r/pics, r/Jokes), while some others are much more homogenous (e.g., r/TwoXChromosomes, r/Fitness, r/gaming, r/Conservative). Also, some subreddits are very strict about enforcing their rules, which does not necessarily limited only to toxic content and usually results in a lot of removed or moderated posts and comments (e.g., r/science), which is not the norm in other subreddits.

Second, the subscriber count does not always correlate with the number of posts and comments in that subreddit. For example, r/announcements is the largest subreddit based on subscriber count (154 million) by far, but only Reddit admins can post on that subreddit (which they only do once every few months), even if any user can still comment on each post. As a result, this subreddit is much less active and has fewer posts and comments than other popular subreddits outside of the top-100 with around 3 million subscribers like r/soccer, r/cars, or r/learnprogramming. One way to check for the number of posts and comments within each subreddit is by using the pushshift API, which was designed and created by the r/datasets mod team. Another way is using https://subredditstats.com/, where it provides several subreddit stats such as posts, comments, growth, posts/comments/day, etc. Unlike Twitter, where all users are practically playing in a single large pool, Reddit is very much different. Each subreddit is somehow unique, and most users are likely to subscribe to only a few subreddits. As such, using user as the only unit of analysis will be justified in Twitter's case but not in Reddit's case, where subreddit is as important a unit of analysis as (if not more important than) user.

That being said, my recommendation would be this:

First, please provide the complete list of those 100 subreddits. This information is critical for understanding the context of the findings in this study.

Second, please keep an eye out for differences in toxicity levels across all 100 subreddits. If the gaps between them aren't too wide, the authors may be able to generalize their findings to other similar large subreddits, though not the entire Reddit. However, if the gaps are so large, it would be prudent to first group those subreddits based on their toxicity level.

Third, once the context has been established as a result of the preceding second phase, the authors may then continue their analysis at the user level, as they did in this current manuscript.

To summarize, analyzing the top 100 subreddits based on subscriber count is acceptable, and the results will undoubtedly be useful. However, generalizing Reddit (or Redditors) as a whole based on these 100 subreddits is not justified.

Validity of the findings

The authors mentioned the use of crowdsourcing to label their initial dataset on 10,000 comments, but apart from the definition of "toxic behavior" on line 94, there is no mention of what define a post or a comment as "slightly toxic" and "highly toxic" in the labeling process in this study. One good example can be seen on the Jigsaw/Conversation AI team's project (https://www.kaggle.com/c/jigsaw-unintended-bias-in-toxicity-classification/data), in which they define "toxic" as "a rude, disrespectful, or unreasonable comment that is somewhat likely to make you leave a discussion or give up on sharing your perspective" and "very toxic" as "a very hateful, aggressive, or disrespectful comment that is very likely to make you leave a discussion or give up on sharing your perspective". Using the example of Figure 1, would that comment still be considered toxic (slightly? or highly?) if there was no f-word profanity while keeping all other words intact? The answer to this question is highly relevant, since the existence of profanity word is one of the key determiner or toxic comment in Detoxify, a popular open-source Python library for identifying potential toxic comments online has this very same issue, as specified by its authors.

If the authors had not established a specific definition beforehand, it is possible that each Figure-Eight worker who labeled the comments had their own subjective definition. As such, it may have introduced bias, particularly identity bias against minority groups, despite the high level of consensus among labelers. I am aware that repeating the labeling process at this stage may be challenging, but the labeling process is crucial to the validity of the findings. Therefore, additional approaches are required to ensure that this potential bias is addressed.

Some references that might be useful regarding this identity bias issues in toxic content detection:

Vaidya, A., Mai, F., & Ning, Y. (2020, May). Empirical analysis of multi-task learning for reducing identity bias in toxic comment detection. In Proceedings of the International AAAI Conference on Web and Social Media (Vol. 14, pp. 683-693).
https://ojs.aaai.org/index.php/ICWSM/article/view/7334

Hanu, L., Thewlis, J., & Haco, S. (2021). How ai is learning to identify toxic online content. Scientific American, 8.
https://www.scientificamerican.com/article/can-ai-identify-toxic-online-content/

Zhao, Z., Zhang, Z., & Hopfgartner, F. (2022). Utilizing subjectivity level to mitigate identity term bias in toxic comments classification. Online Social Networks and Media, 29, 100205.
https://doi.org/10.1016/j.osnem.2022.100205

Additional comments

Still related to profanities, some large SFW subreddits literally have profanities as part of their names (e.g., r/tifu, r/interestingasf***, r/WTF, r/nextf******level), which may result in profanities to be commonly found in the posts and comments. I was wondering if that would mean these subreddits will automatically get higher toxicity levels even if the tone of the posts/comments is not hateful?

As some studies suggested, the use of profanities in a text does not always mean it is hateful nor hostile. Even more so when they are used among friends instead of strangers. It would be nice to see if the authors have considered this in their analysis.

See:
Madukwe, K. J., & Gao, X. (2019, December). The thin line between hate and profanity. In Australasian Joint Conference on Artificial Intelligence (pp. 344-356). Springer, Cham.
https://link.springer.com/chapter/10.1007/978-3-030-35288-2_28

Radfar, B., Shivaram, K., & Culotta, A. (2020, May). Characterizing variation in toxic language by social context. In Proceedings of the International AAAI Conference on Web and Social Media (Vol. 14, pp. 959-963).
https://ojs.aaai.org/index.php/ICWSM/article/view/7366/7220


Last but not least, please review the manuscript for typographical errors. For example, "we" should have been "We" on line 263 because it is the start of a new sentence.


I applaud the authors for their massive dataset and intriguing research article. I hope the authors find my comments useful in improving the soundness of their research and the validity of their findings.

Reviewer 2 ·

Basic reporting

I suggest the authors consider removing the Background heading and place the first four subheadings right under the Introduction. The Introduction section already explains the study’s justification and thus serves as the background, too. If the authors want to keep the Background section separate from the Introduction, I would suggest using either “Related Works” or “Literature Review” subheading.

The last paragraph in the “Toxic Behavior in Online Communities” section (line 151-154) is rather redundant with the “Research Questions” section. I suggest removing either one of them.

I believe the “Data Collection Site” is more appropriate to be placed in the Methodology section.

The third paragraph in the “Obtain Corpus” section (line 203-211) is somewhat perplexing. I tried doing the math and the results did not add up. Please make it clear.

Typo in line 263, it should be “We” not “we”.

Experimental design

Why did the authors only make two rather extreme classifications? What about posts/comments that fall into the neither category i.e., moderately toxic? I believe the authors need adding more justification why they leave the moderate category from the table.

Why did the authors limit themselves to two rather extreme classifications? What about posts/comments that do not fall into either category (i.e., moderately toxic)? I believe the authors should provide more justification for leaving the moderate category out of the table.

How many workers rated each comment/post? How many comments/posts did each worker rate? I noticed the authors used Gwet’s gamma coefficient but was not sure with the number of workers/raters they had for every post/comment and the numbers of posts/comments/ that the workers/raters rated. Please mention them in the manuscript.

Did the raters/workers get something in return for participating in this study? If so, please describe it, if not, please state the justification for their “free” or voluntary work.

Validity of the findings

I believe the raters played critical roles in the study as the post/comment classifications derived from their work. However, I was surprised that I could not find the description of the raters’ demographics. People of different gender, race, ethnicity, and other demographic variables, may have different standards for judging toxicity level of a comment/post. The authors should mention this potential bias in the manuscript and explain it as one of the study’s limitations.

I was also wondering how the authors ensured that the raters did their job well. Did the raters receive any brief training, such as being shown classification examples? Did the raters receive any feedback for their work?

In determining users’ toxicity (line 297-299), I was curious why the authors treat the highly-toxic and slightly-toxic comments/posts as if they were similar? I expected there would be different scores for different levels of toxicity. If they had the same score, why did the authors classify them into different categories in the first place?

Reviewer 3 ·

Basic reporting

The paper is interesting and easy to follow, but proofreading would be helpful.

The introduction and related work give a good understanding of the previous work and the need for such work.

The data and models are publicly available.

Experimental design

Good. Please see the comments below.

Validity of the findings

Good. Please see the comments below.

Additional comments

What are the rules that the authors and labelers use to define a post or comment as toxic or not? Or slightly toxic or highly toxic? A topic could be toxic for a community or a certain period but not for another community, so how do the authors deal with such cases?

Page 5: This link http://redditmetrics.com/top can’t be reached.

Page 5: “Using the combined bot list and Pushshift API, we removed additional bot accounts.” How many were removed?

Page 5: “we targeted the top 100 subreddits with the highest subscriber count.” What were those 100 subreddits?

Page 5: I am not sure Figure 2 adds much to the readers since this is a standard applied ML problem.

Page 6: How many workers labeled the dataset? Did the authors compute the agreement at two levels (i.e., (1) slightly toxic or (2) highly toxic)?

Page 7: “For instance, if a user u created three posts, one labeled highly-toxic, one slightly-toxic, and one non-toxic, this user u is determined as 67% toxic.” What about other cases? What about some posts that the labelers could misinterpret? Or posts that had some mistakes and resulted in a misunderstanding of the posts and got labeled as toxic?

Please provide the details of all the preprocessing. For example, what feature transformation was done? How was the class imbalance addressed? What parameters were used for the different models?

Page 8: Did you use all features separately or concatenate them?

Page 9: Auc should be AUC.

Please report and comment on the precision and recall for all models.

What are the important features based on the random forest?

Given the possibility of errors in labeling, the classification results of the classic machine learning models are not very good. The highest F1 score is 67%.

The paper has some tables (e.g., 6, 7, 8, 9) that could be converted to figures to facilitate the comparison and increase the understating.

Page 11: “To avoid any issues that might arise by posting a single toxic post, any user who shows a change in toxicity due to a single post was removed.” What if two posts? What if some users have changed their behavior from non-toxic to toxic and then from toxic to non-toxic? Do you consider this shift as one or two changes? What if this shift happened several times during the 15 years of the collected data? Do you count them once in Table 5 or multiple times? Also, can you show the counter for each of the three conditions? Something like a histogram would be useful. I am trying to know if this change happened once, twice, or several times. If it is once or twice, then maybe it is due to a single mistake from the user (not a behavior) or an error in labeling the data; otherwise, it could be a serious problem. If you could show some examples of those cases, that could be helpful for the readers.

Page 11: Can you visualize the change in the toxicity of users’ posts over the 15 years? Can we know when exactly those happened? Did they happen recently or years ago? And was there any reason for this shift (e.g., political, economic, etc.)?

Page 11: “engaging with multiple communities can cause users to exhibit changes in their toxic posting behavior.” This could be an interesting result. For those users, can you please show how many subreddits were they posting or commenting to?

Page 12: “at at” remove one

Were the negative and positive files used for classification or sentiment analysis? https://github.com/Hind-Almerekhi/toxicPostingBehavior/tree/main/Data

---

## Round 0.2 · accepted · Accept

I commend the authors for the thorough revisions performed following reviewer comments. All three reviewers and me agree that the submission can now be accepted for publication in its current form.

Reviewer 1 ·

Basic reporting

The revised manuscript addresses the issues with post and comment definitions appropriately.

Experimental design

The authors have clarified their experimental design. Although it differs from my recommendation from the previous round of review, their decisions are fully justified.

Validity of the findings

The authors have added the missing information that may have affected the validity of their findings in the previous version of the manuscript. They also added another step of manual validation on random samples of their dataset. Additionally, the limitations of their methods and findings have been properly stated.

Additional comments

No comment.

Reviewer 2 ·

Basic reporting

The authors have implemented my suggestions.

Experimental design

The authors have addressed all of my concerns and comments.

Validity of the findings

It is unfortunate that demographic information about the raters was not available. I was, however, aware that it was beyond the authors' capabilities. I am nevertheless pleased that the authors clarified this issue.

Additional comments

I have no additional concerns or comments.

Reviewer 3 ·

Basic reporting

no comment

Experimental design

no comment

Validity of the findings

no comment

Additional comments

I would like to thank the authors for clarifying many points and addressing all my comments.